

# Constructing a data-driven receptor model for organic and inorganic aerosol - a synthesis analysis of eight mass spectrometric data sets from a boreal forest site

Mikko Äijälä[1], Kaspar R. Daellenbach[1], Francesco Canonaco[2], Liine Heikkinen[1], Heikki Junninen[1,3],
Tuukka Petäjä[1], Markku Kulmala[1], André S.H. Prévôt[2], and Mikael Ehn[1]

[1] Institute for Atmospheric and Earth System Research / Physics, University of Helsinki, Helsinki, Finland
[2] Laboratory of Atmospheric Chemistry, Paul Scherrer Institute, Villigen, Switzerland
[3] Laboratory of Environmental Physics, University of Tartu, Tartu, Estonia

*Correspondence to*: Mikael Ehn (mikael.ehn@helsinki.fi)

**Abstract.** The interactions between organic and inorganic aerosol chemical components are integral to understanding and modelling climate and health-relevant aerosol physicochemical properties, such as volatility, hygroscopicity, light scattering and toxicity. This study presents a synthesis analysis for eight data sets, of non-refractory aerosol composition, measured at a boreal forest site. The measurements, performed with an aerosol mass spectrometer, cover in total around 9 months over the course of 3 years. In our statistical analysis, we use the complete organic and inorganic unit-resolution mass spectra, as opposed to the more common approach of only including the organic fraction. The analysis is based on iterative, combined use of (1) data reduction, (2) classification and (3) scaling tools, producing a data-driven chemical mass balance type of model capable of describing site-specific aerosol composition. The receptor model we constructed was able to explain $83 \pm 8$ % of variation in data, increased to $96 \pm 3$ % when signals from low signal-to-noise variables were not considered. The resulting interpretation of an extensive set of aerosol mass spectrometric data infers seven distinct aerosol chemical components for a rural boreal forest site: ammonium sulphate (35% of mass), low and semi-volatile oxidised organic aerosols (27 and 12%), biomass burning organic aerosol (11%), a nitrate containing organic aerosol type (7%), ammonium nitrate (5%), and hydrocarbon-like organic aerosol (3%). Some of the additionally observed, rare outlier aerosol types likely emerge due to surface ionisation effects, and likely represent amine compounds from an unknown source and alkaline metals from emissions of a nearby district heating plant. Compared to traditional, simplistic inorganics apportionment methods for aerosol mass spectrometer data, our statistics-based method provides an improved, more robust approach, yielding readily useful information for the modelling of submicron atmospheric aerosols physical and chemical properties. The results also shed light on the division between organic and inorganic aerosol types and dynamics of salt formation in aerosol. Equally importantly, the combined methodology exemplifies an iterative analysis, using consequent analysis steps by a combination of statistical methods. Such an approach offers new ways to home in on physicochemically sensible solutions with minimal need for a priori information or analyst interference. We therefore suggest that similar statistics-based approaches offer significant potential for un/semi supervised machine-learning applications in future analyses of aerosol mass spectrometric data.



## 1 Introduction

Along with particle size, aerosol chemical composition is fundamental in understanding aerosol physicochemical effects such as hygroscopicity, volatility, optics and toxicity (Bilde et al., 2015; Swietlicki et al., 2008; Zimmermann, 2015). In the past decade aerosol mass spectrometry has provided a way to quantitatively resolve basic chemical composition of aerosol in almost

real-time. This not only enables basic chemical speciation into organic and common inorganic ion species, but also produces a wealth of complex mass spectrometric data. It has since become clear that these data sets, although superficially hard-to-interpret, are rich in chemical information and their statistical analysis yields considerable new knowledge. However, tapping into this information source requires use of advanced statistical tools and chemometric methods. Consequently, advanced statistical methods for data reduction have quickly gained traction in aerosol mass spectrometry, and are presently widely used

for deconvolution of complex, organic mass spectra into their underlying components. Specifically, the Positive Matrix Factorization algorithm (PMF, Paatero and Tapper, 1994) has achieved a predominant status as the state-of-the-art analysis tool for deconvolving aerosol mass spectrometric data. Factorisation methods such as PMF notably allow for the condensation of information found in high-dimension data matrices into a manageable number of factors, corresponding to e.g. aerosol chemical species, sources or processes. Data reduction often additionally allows for improved visualisation, aiding in

interpretation of the underlying aerosol chemical phenomena.

In exploratory factor analysis, the principal difficulties often relate to deciding the optimal number of factors, choosing between multiple solutions of mathematically similar quality, and estimating the reliability and uncertainty of the results. Lacking robust but easy-to-use mathematical tools, the selection and interpretation of factorisation solutions remains prone to subjective bias by the analyst. Evaluation and verification of a factorisation solution thus generally requires meticulous study and

understanding of e.g. correlations with auxiliary data, temporal changes and cycles and spectral references. While statistics-driven methods for spectra comparison and classification as of yet remain marginal in aerosol mass spectrometry, they do show promise in their capability to automatically group similar spectra based on their chemically relevant features, producing comparable classifications to those performed manually by expert analysts (Äijälä et al., 2017; Rebotier and Prather, 2007; Freutel et al., 2013).

The overwhelming majority of PMF analyses to date from AMS have been performed on the organic fraction alone (Zhang et al., 2011). Contrary to popular belief, there exists no tenable reasons to limiting chemometric analysis to organic signals, as exemplified by the analyses of Sun et al. (2012) and Hao et al. (2014). Although it requires some additional data preparation and processing, inclusion of inorganics provides additional insight into e.g. salt formation in aerosol. In this work, we apply data reduction and classification methods for analysing organic and inorganic aerosol mass spectral data from several

measurement campaigns in the boreal forest. We then derive a comprehensive receptor model resolving the dominant aerosol categories at the site. In addition, by presenting an example of a semi-supervised, statistics-driven analysis of large mass spectral data sets, we hope to pave the way for machine learning based data analysis approaches, reducing the need for expert analyst input and subjective judgement at each step.



## 2 Methods

Our instrumentation, data processing, measurement site and analysis algorithms have been conscientiously described in previous literature. Thus, we focus on the new aspects of this work, showing how the individual methods can be connected to form an analysis chain, and to exemplify how chemometric information can be propagated through it. In short, we will first

cover the measurement site, SMEAR II and the sets of data available to us (Section 2.1). We then describe our mass spectrometer instrument and preparation of data (Sect. 2.2). In Section 2.3, we will briefly go through the various statistical tools and algorithms, covering the basics of data factorisation, classification of spectra using a clustering algorithm, clustering solution evaluation, and detail the pre and post-weighting involved. Section 2.4 describes typical reference methods for inorganics and nitrate apportionment: an ion balance scheme and a separate parametrisation for estimating organonitrate

loading, to provide a comparison for the inorganic speciation from our statistics-based receptor model. Finally, in Section 2.5, we present a summarised, step-by-step description of how the methods were combined to produce a receptor model for aerosol composition at the measurement site.

### 2.1 Measurement site and collection of data

### 2.1.1. The SMEAR II site

The AMS data of this study was collected at the SMEAR II site (Station for Measuring Ecosystem-Atmosphere-Relationships) in Hyytiälä, Southern Finland (61°50'40"N, 24°17'013"E). The site is a well-known and equipped atmospheric research station, representing rural, background atmosphere in the boreal forest biome. The site and earlier measurements therein have been extensively described and reported in literature (e.g. Hari and Kulmala, 2005; Williams et al., 2011; Äijälä et al., 2017). The environment consists mostly of Scots pine (Pinus Sylvestris) dominated forests - 90 % of land in the nearest 50 km, and

94 % in the nearest 5 km is forested (Williams et al., 2011). Anthropogenic aerosol sources in the region include the towns Orivesi (pop. 9500; 19 km south) and Tampere (pop. 213 000; 48 km south-west), as well as two sawmills and a pellet factory in the village of Korkeakoski, Juupajoki (7 km east-south-east of the station). The surrounding countryside is sparsely populated (5 to 10 inhabitants / $km^2$), and although emissions from agriculture, traffic, domestic heating, and cooking and other combustion sources (saunas, barbeques, agricultural machinery etc.) are limited, they are clearly observable at the station,

and may increase aerosol loading in often plume-type pollution events. The anthropogenic organic aerosols were further analysed previously (Äijälä et al., 2017). However, much of the aerosol loading is attributable to regional biogenic secondary organic aerosol (SOA; Corrigan et al., 2013; Crippa et al., 2014; Allan et al., 2006) and long-range transport from industrial regions in Southern Finland, Western Russia and Central Europe (Kulmala et al., 2000; Patokoski et al., 2015; Niemi et al., 2009; Sogacheva et al., 2005).



### 2.1.2 Data sets

In this study, the aerosol composition was monitored by an AMS between to 2008 and 2011, during several short measurement campaigns. Notable larger, intensive campaigns at the time were the EUCAARI project (2008-2009; Kulmala et al., 2009; Kulmala et al., 2011) and HUMPPA-COPEC (2010; Williams et al., 2011; Corrigan et al., 2013). The sets of data used along

with their timeframes are shown in Table 1.

**Table 1. Data sets used in this study and their timeframes.**

| Data set number | Data set name | Campaign | Start time | End time |
|---|---|---|---|---|
| I | "May 2008" | EUCAARI | 29.4.2008 | 8.6.2008 |
| II | "Sep 2008" | EUCAARI | 10.9.2008 | 15.10.2008 |
| III | "Mar 2009" | EUCAARI | 4.3.2009 | 6.4.2009 |
| IV | "May 2009" | | 29.4.2009 | 28.5.2009 |
| V | "Jun 2009" | | 12.6.2009 | 8.8.2009 |
| VI | "Aug 2009" | | 13.8.2009 | 19.9.2009 |
| VII | "Summer 2010" | HUMPPA-COPEC | 9.7.2010 | 7.8.2010 |
| VIII | "Winter 2010" | | 10.11.2010 | 7.1.2011 |

## 2.2 Instrumentation, data processing and preparation

### 2.2.1 The aerosol mass spectrometer (AMS) instrument and basic data processing

The mass spectrometric data for this study was acquired with a Time-of-Flight Aerosol Mass Spectrometer (ToF-AMS), developed by Aerodyne Research Inc. (Billerica, MA, U.S.). AMS instruments in general have been described by Canagaratna et al. (2007), and the compact ToF analyser version (CToF) used in this study by Drewnick et al. (2005). Additional, more specific details related to the specific instrument we used are available in our previous study (Äijälä et al., 2017).

In brief, the AMS instrument sucks sample aerosol from atmospheric pressure to vacuum conditions through an inlet system

consisting of a critical orifice and a particle concentrating aerodynamic lens (Liu et al., 2007). The sample aerosol beam is directed at a vaporizer operated at 600 degrees Celsius, whereby flash vapourisation of non-refractory aerosol components occurs. The resulting vapour is ionized using 70 eV electron impact ionisation – a well-characterised hard ionisation technique, resulting in rather universal and predictable but highly fragmenting ionisation. Finally, the ions are led to an orthogonal extraction reflectron time-of-flight mass analyser, where the ions' mass-to-charge (m/z) ratios are measured.

The per-amu (atomic mass unit) analyser signal is subsequently quantified based on instrument response calibrations and corrections (among others the correction for relative ionisation efficiency between the species; RIE; Allan et al., 2004); supplementary information Sect S.4). Individual, unit-mass-resolution amu signals are then chemically speciated, based on



chemical information on fragmentation and air composition (see Allan et al., 2003b), for details). Additional, specific minor modifications to our instrument have been discussed in our previous work (Äijälä et al., 2017).

### 2.2.2 Data preparation and down-weighting

After basic processing, the data was further prepared, to serve as input for factorisation (described in following Section, 2.3).
The organic and inorganic data and related uncertainties were extracted, and down-weighting of signals performed. The procedure for extraction and preparation of AMS organic signal and related error matrices has been described by Allan et al. (2003b) and Ulbrich and co-workers (2009).

In short, measurement points or variables with missing data were omitted and error matrices calculated, based on a function accounting for both counting statistics induced uncertainty as well as background noise from the detector and electronics. The
signals were then down-weighted by multiplying the error matrix conveyed uncertainty values for low signal-to-noise ratio (SNR) variables with a scalar: "weak" variables (SNR < 3) were down-weighted by a factor of 2 and "bad" variables (SNR < 1) by 10. The procedure for inorganics ($SO_4$, $NO_3$, $NH_4$, Chl; i.e. sulphates, nitrates, ammonia and chloride species) was similar to that used for the organics ("org"), including for the down-weighting of signals derived from fragmentation calculations. Analogous to the basic procedure of down-weighting "duplicate information" organic signals, e.g. those derived from $m/z$ 44
Th (mainly $CO_2^+$), similarly derived inorganic signal weights were normalized so that their weight of the original plus "duplicate" signals equalled that of the original signal. Finally, the matrices for all the ion species (org, $SO_4$, $NO_3$, $NH_4$, Chl; in nitrate equivalent mass), were combined to form the final input matrices for factorisation, while retaining speciation information in the ion indexing.

### 2.3 Statistical methods and algorithms

**2.3.1 Positive matrix factorixation**

For factorisation, we used the Positive Matrix Factorisation (PMF) model developed by P. Paatero and colleagues (Paatero, 1997, 1999; Paatero and Tapper, 1994), and widely used for analysis of AMS data since 2007 (Lanz et al., 2007b; Zhang et al., 2011). In brief, PMF is a statistical model, typically resolving a bilinear linear combination of factor profiles ($G$) and time series ($F$) best describing the measured data matrix ($X$; Equation 1). The residual matrix $E$ then denotes the portion of data left
unexplained by the model (i.e. residual). PMF model is thus formulated:

$$X_{(t \times v)} = G_{(t \times f)} \times F_{(f \times v)} + E_{(t \times v)}. \tag{1}$$

The brackets indicate matrix dimensions, with $v$ denoting number of variables, $t$ the number of time points, and $f$ the number
of factors. As shown in Equation 1, the model can be solved for any $f$ ($< v, t$), requiring it to be selected by the analyst.





The main features setting PMF apart of other similar factorisation models, and making it particularly suitable for atmospheric aerosol models, are on one hand the limitation of factor profiles and time series to positive values and hence drastically reducing the amount of rotational ambiguity. On the other hand, the improved error model where the quantity to minimise is the weighted (typically the measurement uncertainty) residual, resulting in higher weight for the variables with better SNR. In

PMF, the minimum weighted residual is solved using one of the related algorithms, i.e. PMF2 or Multilinear Engine 2 (ME2; Paatero, 1999). Of the two algorithms, ME2 can take in additional equations defined by the user, i.e. constraints the solutions need to adhere to. For running the PMF and ME2 algorithms, we used the Igor Pro (Wavemetrics Inc.) based SoFi (v. 4.8) user interface developed by F. Canonaco and co-workers at Paul Scherrer Institute (PSI). The interface allows input of the pre-processed data and user selected parameters, and calls on the solver algorithms (PMF2 or ME2, depending on assignment) to

return a solution to be displayed and analysed in SoFi (Canonaco et al., 2013).

When PMF is used as a standalone method for source attribution, the selection of solution needs to be carefully validated. Sensitivities towards different number of factors, rotations and initialisation seeds are meticulously analysed, and correlations with auxiliary data computed. A case is then made for why the selection is the best possible. Contrarily, in our analysis approach, we *do not* claim to arrive at optimum solutions for *individual* PMF/ME-2 runs. Instead, we rely on multitude of data

de-convolution runs to uncover the main structures in the ensemble of all data sets, and use statistical classification methods to evaluate the general outlook and commonalities between PMF/ME-2 factors at each analysis phase. As discussed in Section 2.5, this trade-off instead enables us to concentrate on best modelling the entirety *of all data sets*.

### 2.3.2 Relaxed chemical mass balance model

To harmonise the description of aerosol components, we constructed a constrained receptor model, where all the profile

components were constrained. For this purpose we applied a ME-2 based, chemical mass balance (CMB) type of model. CMB models are typically used as receptor models for cases where source profiles are known, and only the mass loading information needs resolving (Friedlander, 1973; Gordon, 1988; Hopke, 1991; Miller et al., 1972;Hopke, 2016). In such mass conservation-based models, the observed loadings are modelled as a sum of multiple individual sources. Although CMB is mathematically often presented as sum of loadings (supplementary information, hereafter also S.I; Sect S.1, Eq. S.1), it can also be thought of

as a special case of the bi-linear model described in Equation 1. Only now the profile matrix ($F$) is assumed fixed, simplifying the problem to resolving the loading matrix ($G$) which minimises the residual ($E$). CMB can be run using the SoFi interface, using the same ME-2 solver as for PMF and ME-2 applications (Canonaco et al., 2013).

In this work, we use a relaxed CMB–like bilinear model (henceforth abbreviated as r-CMB), where all the source profiles are constrained, but allowed to vary within narrow limits. In strict technical terms this approach could be labelled "an extremely

constrained ME-2 model", but we choose to use the term "relaxed CMB" to differentiate between the typical use of ME-2 or constraining only part of the profiles, which allows the model considerably more freedom. We regard our use of the model is much closer to the idea of constraining all profiles than (semi-)exploratory factorisation typical for ME-2. The naming also serves to better highlight the conceptual differences between models in the different analysis phases.





Generally, the biggest problems of the CMB models relate to the selection of source profiles, typically from spectral libraries, and handling of their uncertainty. In our use, the anchor spectra as well as the limits for their allowed variabilities are experimentally derived from data, alleviating some of these typical concerns.

### 2.3.3 k-means clustering

For spectra classification, we selected the k-means algorithm, specifically because in our previous tests it was successful in classifying similar spectral data. The earlier tests additionally yielded of useful information on selection of the dissimilarity metric, as well as algorithm initialisation types and data weighting (Äijälä et al., 2017). K-means (e.g. Ball and Hall, 1965; MacQueen, 1967; Steinhaus, 1956; Jain, 2010)) is a rather simple, iterative algorithm that partitions a group of objects to a predesignated number of groups or 'clusters' based on their relative distances (i.e. dissimilarities). For each iteration, the

algorithm assigns all objects to their closest centroids, which are then re-calculated from the mean variable values of the objects in the updated cluster. The aim is to minimize the within-cluster sum of distance (variance) ($J$) between the objects' ($C_n$) locations ($x_i$) and the cluster centroid $\mu_n$ they are assigned to (Eq. 2):

$$J(C_n) = \sum_{x \in C_n} \|x_i - \mu_n\|^2. \tag{2}$$

The k-means algorithm iteratively converges on (any) minimum of total $J\ (C)$ obtained by summing over all objects $C_n$. To increase chances of finding a global minimum, repetitions using different initialisations are used. Specifically, we used the improved stepwise initialisation 'kmeans++' (Arthur and Vassilvitskii, 2007); available in e.g. Matlab v. 2017a; Math Works Inc., Natick, MA. U.S.).

**2.3.4 Spectral similarity and mass scaling**

Based on our earlier metric comparison (Äijälä et al., 2017), we used (Pearson) correlation as a metric for spectral dissimilarity (or "distance", d; Fortier and Solomon, 1966; Mcquitty, 1966):

$$d(u,v) = 1 - \frac{\sum_{i=1}^{n}(u_i - \bar{u})(v_i - \bar{v})}{\sqrt{\left(\sum_{i=1}^{n}(u_i - \bar{u})^2\right)}\sqrt{\left(\sum_{i=1}^{n}(v_i - \bar{v})^2\right)}}, \tag{3}$$

where u and v are the spectra in vector form, with *m/z* variables as vector components. $\bar{u}$ and $\bar{v}$ are the arithmetic mean values of *u* and *v*.

In clustering mass spectra, data weighting is often applied. Based on previous tests (Äijälä et al., 2017), we applied mass scaling of variables, advocated by Stein and Scott and others (Stein and Scott, 1994; Kim et al., 2012; Horai et al., 2010),

giving additional emphasis to higher mass signals. This common practice is based on the idea that higher mass fragment ions are more indicative of their parent ions, and thus the original characteristic composition, while smaller fragments can be





produced from a wider variety of molecular fragmentation events. In mass scaling the weighted variables ($\hat{x}$) are calculated by multiplying the original variables ($x$) by mass-to-charge-specific weights ($w$), as presented in Equation 4.

$$\hat{x}_{m/z} = x_{m/z} \times w_{m/z} ; \quad w_{m/z} = (m/z)^{\mathbf{s_m}},$$ (4)

where the scaling factor $s_m$ was optimised for each classification separately (SI; Sect. S.2).

### 2.3.5 Silhouette metric and post-weighting

The optimisation of mass scaling was based on silhouette metric (later also abbreviated as "silh"; (Rousseeuw, 1987), ranging between -1 to 1 and providing a straightforward, quantitative way to evaluate performance of the classification algorithm. The
10 object-specific silhouette value $s_i$, defined as

$$s_i = \begin{cases} 1 - \frac{a_i}{b_i}; \text{ for } a_i < b_i \\ 0; \text{ for } a_i = b_i \\ \frac{b(i)}{a(i)} - 1 \text{ for } a_i > b_i \end{cases},$$ (5)

where $a_i$ corresponds to the mean distance to other objects in the same cluster, and $b_i$ similarly to the mean distance to objects in the nearest neighbouring cluster. A silhouette value close to unity indicates the object is well clustered, while a value close
to zero indicates the classification is uncertain, and the point is likely situated in-between two possible centroids. A negative cluster value is indicatory of possible misclassification. Silhouette values can be calculated for any single cluster as the arithmetic mean of the cluster members' silhouettes, or similarly as a mean over all objects, to evaluate the quality of the clustering solution as a whole.

In order to mitigate the k-means algorithm's known sensitivity to outliers, and to improve handling of between-cluster samples,
we applied a simple post-processing to all cluster centroids and variability calculations: the centroid spectra and variabilities were calculated as *weighted* averages μ, and *weighted* standard deviations ($\hat{\sigma}^2$; Eq. 6) respectively, instead of the normal unweighted values (similar to Äijälä et al., 2017). As weights, we used the object specific silhouette values $s_i > 0$ (Eq. 5):

$$\hat{\mu} = \frac{\sum_{i=1}^{N} s_i v_i}{\sum_{i=1}^{N} s_i}; \quad \hat{\sigma}^2 = \frac{\sum_{i=1}^{N} s(v-\mu)^2}{\sum_{i=1}^{N} s_i}; \quad s_i = \max (s_i, 0),$$ (6)

where $v_i$ are the cluster member objects (spectra) This procedure down-weights likely misclassified objects (silhouette < 0) to zero, and penalises the more uncertain or questionable assignations (low silhouette) compared to the decidedly well-clustered objects (silhouette close to unity). Singleton clusters were omitted from this calculation, and their variability thus left undefined.



## 2.4 Standard approximations for aerosol inorganic speciation and organonitrate

### 2.4.1 Ion balance model for inorganics

Aerosol inorganic chemical speciation is better understood than the organic speciation, due to much lower diversity of the chemical compounds involved. There exist a variety of aerosol inorganic equilibrium models, typically used as modules in

atmospheric meteorological and air quality models. However, performing thermodynamic equilibrium calculations is computationally demanding (e.g Fountoukis and Nenes, 2007), and requires a good deal of auxiliary data on thermodynamic conditions and chemical activities. Due to the complexity of the models and increased data needs, simpler approximations are often used in connection to AMS inorganic speciation. In the following ion-balance-scheme description, we denote the respective AMS ion species molar concentrations in square brackets (e.g. $[NH_4^+]$, $[NO_3^-]$, $[SO_4^{2-}]$)

A typical salt formation approximation used for AMS results is the Hong et al. (2017) ion pairing scheme, used in e.g. aerosol volatility and light scattering models (Hong et al., 2017; Zieger et al., 2015). The Hong et al (2017) scheme is based on similar approximation of Gysel et al. (2007), which in turn is a simplification of the more extensive model by Reilly and Wood (1969). We modified the Hong et al. scheme to additionally allow organonitrate (orgNO$_3$) and speciate any leftover $[NH_4^+]$ as its own class ("excess NH$_4^+$"). The full scheme is available in supplementary material (Section S.3), and a schematic description is

presented in Figure 1.

Briefly, in the scheme we apply, $NH_4^+$ is first combined with $SO_4^{2-}$ to form ammonium bisulphate and/or ammonium sulphate depending on the relative concentrations of $[NH_4^+]$ and $[SO_4^{2-}]$. Any leftover $[NH_4^+]$ then combines with $[NO_3^-]$, until all of $[SO_4^{2-}]$ and $[NO_3^-]$ is fully consumed in forming $(NH_4)_2SO_4$ and $NH_4NO_3$. After this point, any leftover $[NH_4^+]$ is considered "excess" and assigned to a separate class. For comparability with other models, any nitrate not in $NH_4NO_3$ is labelled organic.

Despite the label, we note this class not only encompasses organonitrates, but also any $NO^+$ fragment signal from amines, N-containing organics and may even contain influences of other inorganic nitrate species, such as $KNO_3$, which are not considered separately in this simple model. Finally, since chloride loadings at the measurement site are generally negligible, neutralisation of hydrochloric acid ($H_2O$:HCl) was not included to keep this scheme rather simple. We note ion balance schemes depending on relative ion abundances, such as the one described here, can be sensitive to measurement uncertainties (e.g. errors in RIE

values) of these ratios. The topic is further discussed in supplementary information (Sect S.4)




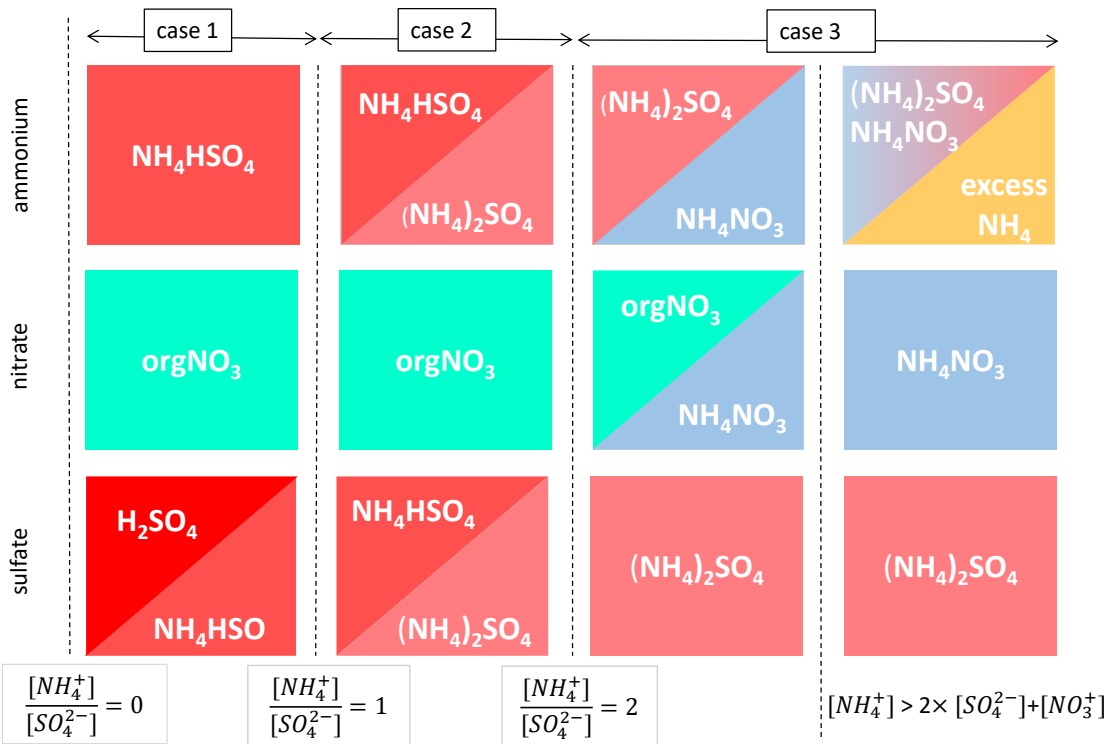

**Figure 1. Schematic representation of the inorganic apportionment scheme. The scheme is divided into three cases according to the ratio of [NH$_4^+$] to [SO$_4^{2-}$]. [NH$_4^+$] first combines with[SO$_4$] to form NH$_4$HSO$_4$ (Case 1), then further to (NH$_4$)$_2$SO$_4$ (Case 2). In these cases, any nitrate observed is considered organic. In Case 3 leftover [NH$_4^+$] then associates with [NO$_3^-$] until all the norganic anions**
**are neutralised. Any leftover [NH$_4^+$] is labelled as "excess NH$_4^+$". Full description of the scheme is given in supplementary material (Sect S.3).**

### 2.4.2 Kiendler-Scharr parameterisation for organonitrate

Organic nitrate estimate in the above model is very sensitive to calibration parameters (see S.I Sect S.4). Therefore, in addition

to the ion-balance based scheme above, we additionally calculated a particulate organonitrate mass estimate (orgNO3$_{mass}$),

based on the nitrate fragmentation ratio-based parameterisation of Kiendler-Scharr et al. (Kiendler-Scharr et al., 2016; Farmer

et al., 2010):

$$orgNO3_{mass} = NO3_{total} \frac{(1+R_{orgNO3})x\ (R_{measured}-R_{calib})}{(1+R_{measured})x\ (R_{orgNO3}-R_{calib})},$$ **(6)**

where R refers to the ratio of nitrate signals at 46 and 30 Th, i.e. $R = NO_3$ (*m/z* 46 Th) : NO$_3$ (*m/z* 30 Th), for organonitrate

("orgNO$_3$"), NH$_4$NO$_3$ calibration ("calib"), and ambient measurement ("measured"), respectively. For the parameterisation,





we applied an ion ratio $R_{calib} = 0.42$, taken as the average of mass spectrum based AN calibrations (S.I Sect S.6). $R_{OrgNO3}$ value of 0.1 was used, based on the estimate by Kiendler-Scharr and co-workers for their observations on organonitrate spectral properties (Kiendler-Scharr et al., 2016).

## 2.5 Constructing a data-driven r-CMB receptor model

As stated in the Introduction, one of the aims of our work was to derive a robust, harmonised receptor model for the measurement site via explorative analysis. Considering the large amount of campaigns during different seasons, resulting in changing aerosol source contributions and mass spectral profiles, factorisation needed to be performed on a per-campaign (data set) basis. However, instead of performing traditional PMF complete with correlation analysis, source validation and the various sensitivity analyses separately, which would be an arduous task even for a single measurement set, we used the large

amount of data sets to our advantage. Instead of optimising individual factorisations, we constructed an r-CMB model applicable to all data sets. A similar task of constructing a semi-exploratory synthesis aerosol model, albeit applying a different methodology, was undertaken and reported by Sofowote et al. (2015).

To derive the anchors and constraints for a synthesis r-CMB model, we analysed the data in three phases (P-I to P-III; Figure 2), each consisting of factorisation, classification and silhouette-based post-weighting of anchor spectra and their allowed

variabilities. In Phases I and II, a fixed number of 10 factors were resolved. This amount of factors was semi arbitrarily chosen, and in our case likely to be somewhat above the optimal amount for most data sets, leading to over-resolved factor solutions. However, unlike in traditional PMF analysis, we can use additional statistical diagnostics and post-processing options available to deal with potential fallout of unrealistic factor splitting (i.e. classification for identifying outliers and post-processing down-weighting or nullifying their influence. Sensitivity to initialisation seed was examined by performing all runs using 10

initialisation seeds, and generally selecting the solution with lowest normalised residual. In rare cases of a physically unrealistic solutions as the one with lowest residual (e.g. only $NH_4$ species in a factor), a higher residual solution was chosen instead. We conclude the solutions were generally insensitive to seed selection, especially for the factors with non-negligible mass contribution.

### 2.5.1 Phase I: anthropogenic aerosols

In phase I, we performed unconstrained factorisation for all the 8 data sets. With 10 factors this resulted in a total of 80 factor mass spectra. We then determined the dominant spectra classes using k-means clustering. To that purpose, we applied optimised mass scaling for improved data structure, and used silhouette diagnostics to evaluate the optimal number of clusters. We identified the known, common anthropogenic aerosol classes from the silhouette-weighted cluster centroids. This is also an approach advocated by Crippa et al. (2014) in their similar work on a synthesis analyses of several data sets.

For a cluster centroid to qualify as an anchor for further phases of our analysis, we applied the following two criteria: (1) The spectra forming the cluster were present in multiple ($\geq 3$) the data sets, (2) The spectra were interpretable chemically, and had adequate support from previous studies in form of literature and/or calibrations. We note that defining what constitutes as



"interpretable" or "adequate support" is inevitably an analyst (subjective) decision, so we endeavour to make our reasoning transparent in the respective discussion sections. Adhering to criterion (1) also means that factors showing up only for 1 to 2 campaigns, due to special conditions (emission, meteorology etc.), are omitted from the final r-CMB model. We will briefly cover some of the more interesting "outlier observations" in Section 3.4. At the end of Phase I, a number of constrained anchor

spectra and within-cluster-variabilities were obtained. In this case, these corresponded to four anthropogenic classes, which will be discussed in more details in the results section.

### 2.5.2 Phase II: biogenic, secondary organic aerosols

Using the anchors and within-cluster-variabilities, we re-ran factorisation as in P-I, except now partly constrained (ME2; 4 of 10 factors constrained using anchors from P-I). In phase II, we focused on analysing the remaining free factors, likely

corresponding to the biogenic, and assumedly more variable factors (Canonaco et al., 2015;Crippa et al., 2014). The procedure for classification, and the selection criteria for the (assumedly) biogenic SOA in this phase was same as in phase I.

Due to the data-driven analysis approach, specifically the constrained factors being selected from phase I, we do not expect major changes between phase I and phase II results. While arguably the methodology could be further developed to constrain the r-CMB components directly from phase I result, phase II of our analysis currently serves several purposes: 1) it should

narrow down the solution space for improved description of the various SOA types, by constraining the anthropogenic, assumedly primary aerosols. 2) Compared to P-I, the allowed solutions are more similar for all data sets in P-II, which reduces the scatter of the factorisation solutions. This reduces the spectral variability (uncertainty) arising from the analysis process itself, allowing us to iteratively converge on more realistic limit values for the constraints. Ultimately, the limits should reflect the actual, natural chemical variabilities within the aerosol types. 3) Similarity of results between successive, un/semi-

constrained phases allows evaluation of stability, reliability and repeatability of the method, so that it is not e.g. overly sensitive to rotational ambiguity or initialisation parameters of algorithms. This is important since the method described here is new, and its robustness needs to be demonstrated, but less so in potential later use.

### 2.5.3 Phase III: final, constrained receptor model

In phase III, we constructed the r-CMB receptor model. In this phase, all the factors were constrained using anchors and

variabilities from the previous phase result. The number of components in the final r-CMB model, in our case 7, was equal to the total number of selected aerosol types in phase II. With these model constraints, we performed runs for each of the 8 data sets separately. Using the resulting 8×7 factor profiles, we determined the likely range of variability for the aerosol types, and calculated final, silhouette weighted reference spectra for the components by performing a final round of clustering.





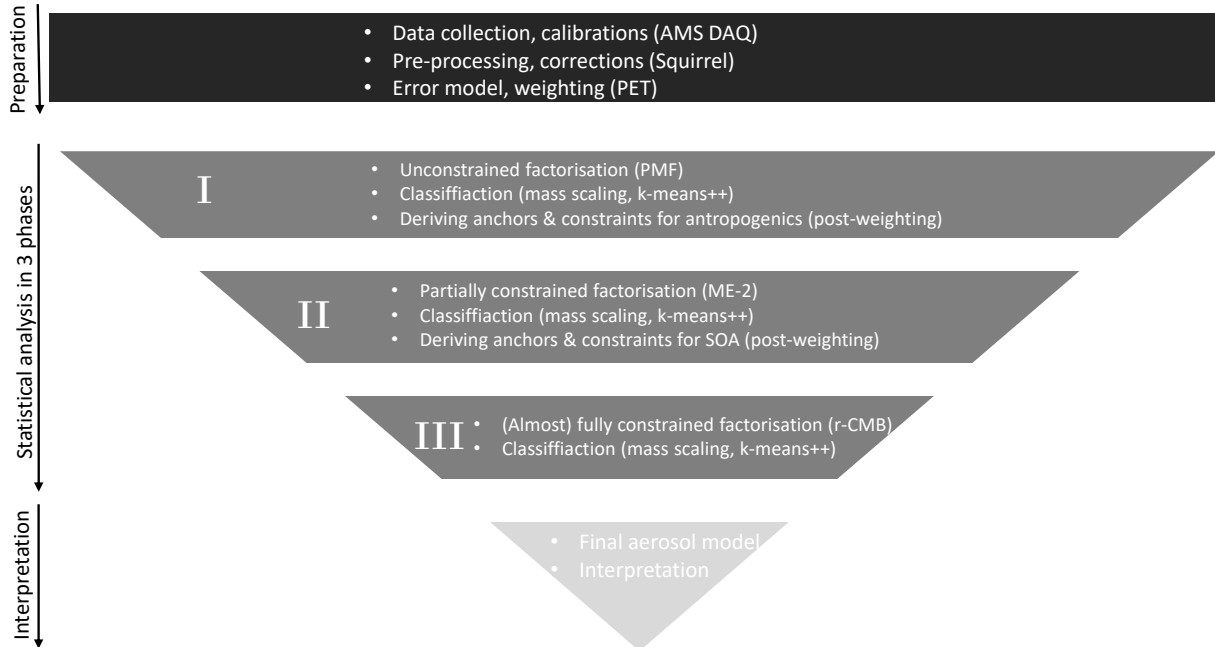

**Figure 2. A flowchart illustrating the analysis using combined methodology. After initial data collection and preparation, statistical analysis is performed in three phases (P-I to P-III). Each phase limits the freedom given to factorisation from completely free (PMF) to nearly fully constrained (r-CMB). Finally, we evaluate and interpret the r-CMB model from an aerosol chemical perspective.**

## 3. Results and discussion

In Section 3.1, we briefly describe the results from analysis phases I to III (P-I to P-III; corresponding to Sections 3.1.1 to 3.1.3), but concentrate more on the receptor model results and their interpretation (Sections 3.2). Finally we will compare our results with reference methods (Section 3.3). Comparison results are available in literature for organic aerosol components (Sect 3.3.1), and in Sect. 3.2 we will compare inorganic speciation with the alternative inorganic attribution methods, described

10 in Methods (Sect 2.4). Finally, we briefly describe some of the outlier observations which contain potentially interesting chemical information (Section 3.4).

When interpreting and identifying aerosol components, we evaluate spectral similarity using the same similarity metric (mass scaled correlation) as for the clustering (Equations 3 and 4). We thus report mass scaled squared correlation coefficients ($r_s^2$) between reference spectra and our corresponding final spectrum for the class (P-III silhouette-weighted centroids; $s_m=1.81$).

15 For easier comparability, all ratios and fractions of signals presented in the following sections are similarly calculated from the corresponding final spectra (P-III).



### 3.1 Receptor model construction steps

#### 3.1.1 Phase I: identification of anthropogenic aerosol components

In phase I, we performed unconstrained PMF runs using 10 factors for all 8 datasets separately. The resulting 80 factor spectra were subsequently clustered. Maximal data structure (silhouette 0.56) was achieved at mass scaling $s_m = 2.12$ for 17 clusters

5 (for details on silhouette analysis, see supplementary material, Sect. S.2 ). The eight clusters with largest population for the phase I solution are shown in Figure 3, and the rest in Section 3.4, where outlier observations are further discussed. Generally, the solutions agreed closely on the largest clusters, lending credibility to the robustness of the approach. The solutions differed mainly regarding outlier classification, which is of secondary importance for our r-CMB model, since outliers are discarded from the model.

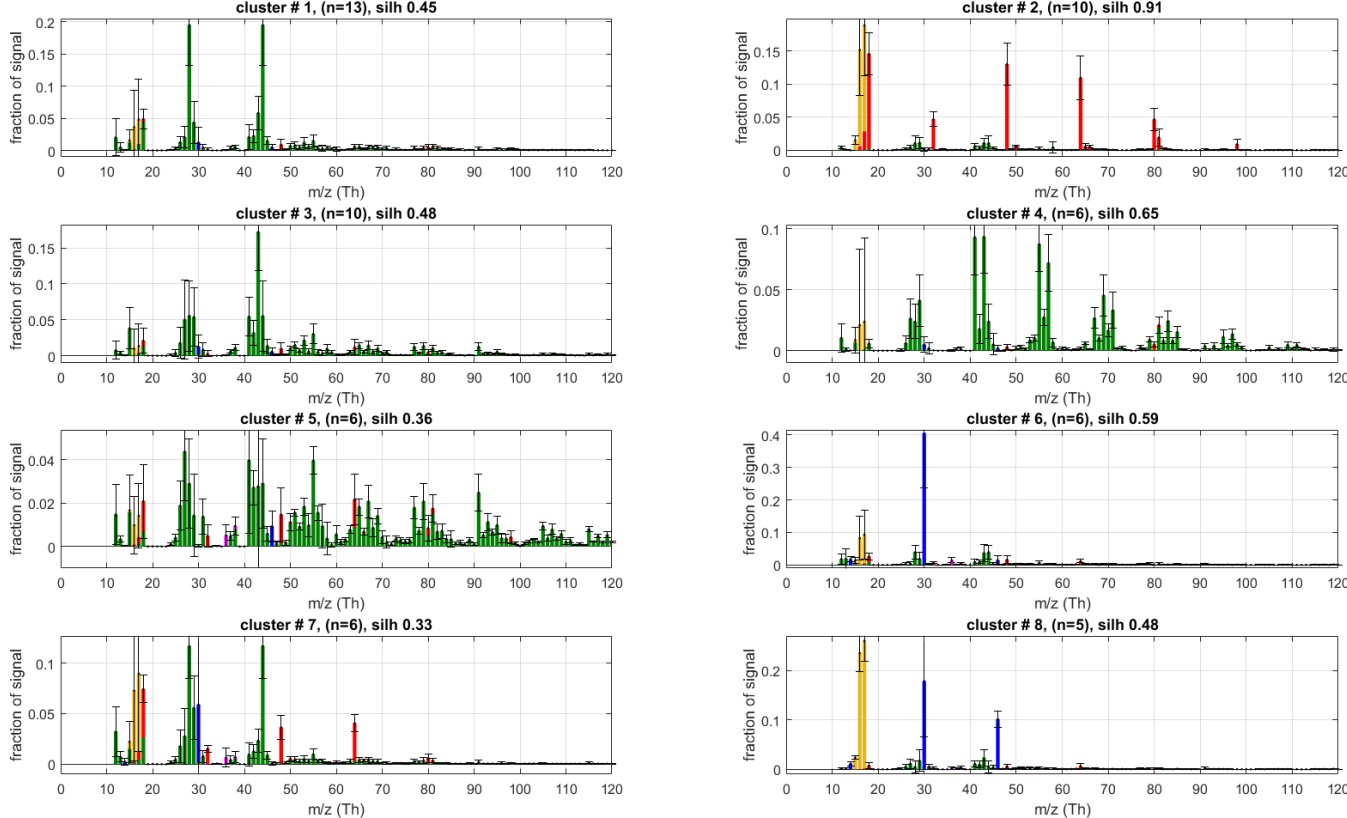

**Figure 3. The 8 largest clusters for P-I classification of factorisation results. Cluster centroids (coloured bars; red: SO₄: blue NO₃; orange: NH₄; magenta: chlorides; green: org) and variabilities (error bars) are silhouette-weighted averages and standard deviations for the cluster members. The main anthopogenic aerosol types were identified as Clusters #2 ('Ammonium sulfate', AS),**
15 **#4 ('Hydrocarbon-like organic aerosol', HOA), #5 ('Biomass burning organic aerosol', BBOA) and #8 ('Ammonium nitrate', AN). Cluster number, silhouette and population (n) are shown in panel titles.**



Unsurprisingly, the classification returns two large clusters of organic resembling the ubiquitous low-volatile oxidised organic aerosols (#1; "LV-OOA) and semi-volatile oxidized organic aerosol ("SV-OOA"; e.g. Aiken et al., 2007; Jimenez et al., 2009; Zhang et al., 2011). Comparing to library spectra, the *m/z* 44 Th ($CO_2^+$) dominated aerosol type (#1) best matches with LV-OOA and OOA-I (Oxidised Organic Aerosol; a historical label corresponding to LV-OOA; (Aiken et al., 2008; Zhang et al.,

2011) spectra from Paris ($r_s^2 = 0.97$; Crippa et al., 2013), Zurich (0.96; Lanz et al., 2007a; Crippa et al., 2013) and Borneo rainforest (0.99; Robinson et al., 2011) as well as the average LV-OOA calculated from 15 Northern Hemisphere datasets (0.94; Ng et al., 2010). Cluster #3 is characterized by high *m/z* 43 Th signal ($C_2H_3O^+$; Aiken et al., 2008), and correlates with SV-OOA and OOA-II (Aiken et al., 2008) spectra from Pasadena (0.74; Hersey et al., 2011), Borneo (0.86; Robinson et al., 2011) and the 15 data set average (0.76; Ng et al., 2010) as well as the laboratory generated SOA spectra generated from

typical pine forest emitted volatile organic compounds (e.g. a-pinene, 0.81; a-terpinene, 0.83; terpinolene, 0.84; (Bahreini et al., 2005)). Abiding by the typical naming convention of AMS derived aerosol types, we label these species LV-OOA (cluster #1) and SV-OOA (#3).

The solution also contains a large cluster (#2) with spectra dominated by ammonium and sulphate ion species. This is in agreement with ammonium sulphate being a main component of ambient aerosols. Although it contains also trace amounts of

other species, we name the $(NH_4)_2SO_4$ –dominated aerosol class (#2) ammonium sulphate ("AS") for brevity.

The main nitrate-containing spectra are divided into two clusters (#6 and #8). The divisive feature seems to be the ratio of *m/z* 46 to 30 Th signals (i.e. $R_{measured}$ in Equation 7), which is much higher in cluster type #8 ($0.44 \pm 0.11$) versus for #6 ($0.08 \pm 0.07$; P-III values; see S.I. Sect S.5 for error estimate). Based on literature we interpret the split to correspond to the division between nitrogen in form of inorganic (ammonium) nitrate (AN), and organic nitrogen, matching with previous AMS

observations (Hao et al., 2014; Farmer et al., 2010; Kiendler-Scharr et al., 2016). The interpretation of cluster #8 as AN is additionally corroborated by its similarity to spectra from pure ammonium nitrate calibration for the instrument, available in supplementary material (Sect S.6). On average, the (Brute-Force Single Particle, BFSP; Drewnick et al., 2015) AN calibrations performed for the instrument yielded an $R_{calib}$ (Equation 7) ratio of $0.49 \pm 0.05$ (mean ± standard deviation), while an MS mode calibration returned an $R_{calib}$ of 0.42. Similarly to naming of the AS class, we use labels organic nitrogen ("ON"; cluster #6)

and ammonium nitrate (AN; cluster #8) for the nitrate-dominated aerosol types. The ON cluster is further discussed in Section 3.3.2. The label ON was chosen to differentiate between the (presumably) organic nitrogen dominated aerosol class (ON), and the part of $NO_3$ ion species deemed likely to be organonitrate ($orgNO_3$).

A fraction of the organic signal observed at *m/z* 44 Th for inorganic salt classes (AS and AN) may be explained by an $CO_2^+$ artefact induced by thermal decomposition of inorganic salts (Pieber et al., 2016). For ammonium nitrate, the ratio of organic

signal at *m/z* 44 Th to total nitrate signal is 2.9% (P-III). Pieber et al. (2016) estimate a contribution of 3.4 %, suggesting most of the organic signal observed in AN may arise from this artefact. This proposition is further discussed in supplementary information (Sect. S.6)

Two of the clusters (#4 and #5) seem related to anthropogenic (primary) organic aerosol types. Cluster #4 has a similar spectrum as the hydrocarbon-like-organic aerosol ("HOA") spectra from the AMS spectral database (Ulbrich et al., 2009), and





closely matches, among others, HOA reported by Zhang et al. (Zhang et al., 2005) for Pittsburgh ($r_s^2 = 0.91$) and the average of de-convolved 15 HOA spectra reported by Ng et al (2010; $r_s^2 = 0.89$). The spectra also exhibits high similarity with traffic emission spectra of diesel bus exhaust (0.86), lubricating oil (0.82) and fuel (0.75), reported by Canagaratna et al. (2004).

Cluster (#5) features high signals for ions typical of biomass burning organic aerosol ("BBOA"; e.g. Alfarra et al., 2007) and

cooking organic aerosol ("COA"; e.g. Mohr et al., 2012). The spectra features the marker signals of levoglucosan (Cubison et al., 2011; Schneider et al., 2006) at $m/z$ 60 ($C_2H_4O_2^+$) and 73 Th ($C_3H_5O_2^+$) along with chloride ions (at $m/z$ 35 and 36 Th) and high fraction of signal at $m/z$ 55 Th ($C_3H_3O^+$; Mohr et al., 2012), pointing to cooking and/or biomass burning emissions. Highest similarities to library spectra (de-convolved via PMF) are found with COA (Mohr et al., 2012; Barcelona; $r_s^2 = 0.70$ and Crippa et al., 2013; Paris; $r_s^2 = 0.59$) and BBOA (e.g. 15 dataset average reported by Ng et al. (2010; $r_s^2 = 0.51$) and BBOA

de-convolved by Crippa et al. (2013; for Paris; $r_s^2 = 0.50$). Similarity to SV-OOA library samples are also moderately high (e.g. Ng et al., 2010; 15 dataset average; $r_s^2 = 0.59$).

The differentiation between HOA versus BBOA or COA can often be resolved even from unit resolution spectra, using the $f55$ to $f57$ ratio (Mohr et al., 2012), and the differences in mass spectral fingerprints higher up on the $m/z$ axis (resolvable using mass scaling; Äijälä et al., 2017). However, the distinction between COA and BBOA aerosol types is much more delicate due

to very high UMR spectral similarity also for higher $m/z$ variables, (e.g. $r_s^2 = 0.79$ for COA and BBOA reported by Mohr et al., (2012). The main difference between the COA and BBOA aerosol types is the absolute level signals from levoglocosan fragments, the quantitative interpretation of which is difficult due to (1) levoglucosan production being determined by combustion temperature (Shafizadeh, 1984), (2) levoglucosan originating both from BBOA and COA (Mohr et al., 2012) and (3) levoglucosan sinks may be considerable in the atmosphere (Hoffmann et al., 2009), which affects especially transported

aerosol. Due to the remote location of the measurement site and general prevalence of BBOA over COA in urban aerosol loadings (e.g. Daellenbach et al., 2017) we conclude BBOA is more likely the dominant component for this aerosol type, so we will use the class label "BBOA" for brevity. Due to high spectral similarity, we find it extremely likely any COA contribution would be apportioned to this class, but without the benefit of high mass resolution data, the convolution seems insolvable at this time.

Cluster 7 spectrum offers little in terms of unique spectral features, and it appears as it could be represented as a combination of the more distinct AS (#2), LV-OOA #1) and ON (#6) aerosol types. It is unclear if this class represents an actual aerosol chemical type, or if it is due to incomplete resolving of the aforementioned species in the PMF model. We note the organics part of AS, LV-OOA and ON are all highly oxidised, which may imply similar level of aging, thus similar origins for these species. Organic spectral component are further analysed and discussed in Sect. 3.2.2.

Based on this interpretation and evaluation of criteria outlined in Sect. 2.5, we decided to select as the main representative anthropogenic aerosol types the following: ammonium sulfate (AS, cluster #2, $n = 10$, silhouette = 0.91) ammonium nitrate (AN, #8, $n = 5$, silh = 0.48), hydrocarbon-like organic aerosol, (HOA, #2, n = 6, silh = 0.65) and biomass burning organic aerosol (BBOA, #5, $n = 6$, silh = 0.36). The silhouette values can be taken to represent separation distance from neighbouring aerosol types. For comparison, silhouette values for some of the anthropogenic organic aerosol types are available in Äijälä et




al. (2017), but to our knowledge no precedent exists for mixed or inorganic aerosols. Generally, the more 'unique' the spectra of a group and the higher within-cluster-cohesion, the higher the silhouette.

### 3.1.2 Phase II: classification of biogenic, secondary organic aerosols

In the second phase of our analysis, ME-2 factorisations were run for ten factors for all the data sets. We constrained 4 out of the 10 factors with the anchors and variabilities for anthropogenic aerosol types, derived from previous phase (AS, AN, HOA, BBOA). The resulting 80 factor profiles were again extracted and classified. The classification solutions featured generally higher silhouette value than in the first phase, which is at least partly explained by constrained spectra being forced to conform to their set limits. The highest total silhouette (0.66) was obtained for 15 clusters (at $s_m$ = 2.41). Again, the inter-solution variability for the solutions inspected was low for the main classes. The phase II solution is available in S.I (Figure S.4). Overall, the solution very closely resembles the result from phase I (Figure 3).

The expected LV-OOA (#1; $n$ = 14; silh 0.64) and SV-OOA (#3; $n$ = 9; silh 0.44) aerosol types again rank among the most typical classifications. Their moderate silhouettes reflect higher variability within these classes, corresponding to results from earlier studies (e.g. Canonaco et al., 2015), and/or closer proximity to neighbouring aerosol types, than for the AN, AS or HOA types. The result may suggest seasonal or other dataset-specific variability for SOA, which supports partitioning the data on a per-campaign basis. In accordance with typical AMS organic aerosol classification conventions laid out by e.g. Aiken et al. (2008), we opt for two classes of oxidised aerosols. We thus select clusters #1 and #3 (P-II) to represent LV-OOA and SV-OOA (Aiken et al., 2008; Jimenez et al., 2009) respectively.

For P-III of our analysis, we additionally fix the organic nitrogen class, (ON, P-II cluster #8). Irrespective of the exact chemical composition and label of this aerosol component, we assess there is enough literature support (among others Kiendler-Scharr et al., 2016; Farmer et al., 2010; Drewnick et al., 2015; Murphy et al., 2007; Hao et al., 2014) for inclusion of nitrogen containing aerosol types other than AN, to warrant the inclusion of this class. In any case, the classification of nitrate signal at $m/z$ 30 Th to a distinct class seems statistically robust, as exhibited by its emergence as a free factor in both P-I and P-II solutions. Due to the importance of nitrogen containing species in SOA composition and formation (e.g. (Kiendler-Scharr et al., 2016; Berkemeier et al., 2016) we find it an important aerosol class to include, examine and further interpret. The mixed cluster #7 also emerges for 4 data sets, but with notably low silhouette (0.18), suggestive of low within-cluster cohesion. As we still lack a distinct chemical interpretation for this class, beyond the hypothesis of incomplete resolution of aged aerosol species in factorisation, we will not include the mixed class (#7) in our final receptor model.

### 3.1.3 Phase III: Final r-CMB receptor model

In the final phase (P-III) of constructing our r-CMB receptor model, we used 7 factors which were all constrained with the profiles and allowed variabilities from the previous phase (P-II, AS, LV-OOA, SV-OOA, BBOA, ON, HOA, AN). The ME-2 algorithm was tasked to resolve the factors' temporal behaviour.





To derive final characteristic spectra for the model components, as well as to study the variability of spectra in the solutions, we once more applied the same clustering procedure and silhouette analysis as for previous phases. The maximal structure (silh 0.85) was achieved for the seven cluster solution ($s_m = 1.81$), which was to be expected considering ME-2 was run with 7 rather strictly constrained factors in this phase. With silhouette weighting applied, we obtain the final spectra and variabilities (Figure 4). We note this final clustering and weighting step mainly serves to provide an estimate of variability within each aerosol type, but also yields final spectra to be used as library references for the outcome of this work. Details of the solution of the r-CMB model are discussed in following sections, from the perspective of mass attribution (Section 3.2.1) and spectral characteristics (Section 3.2.2).

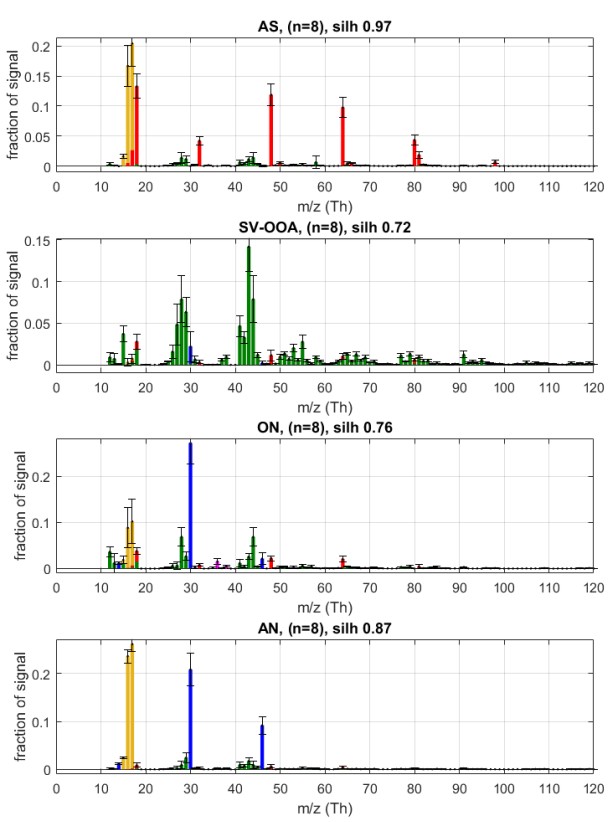

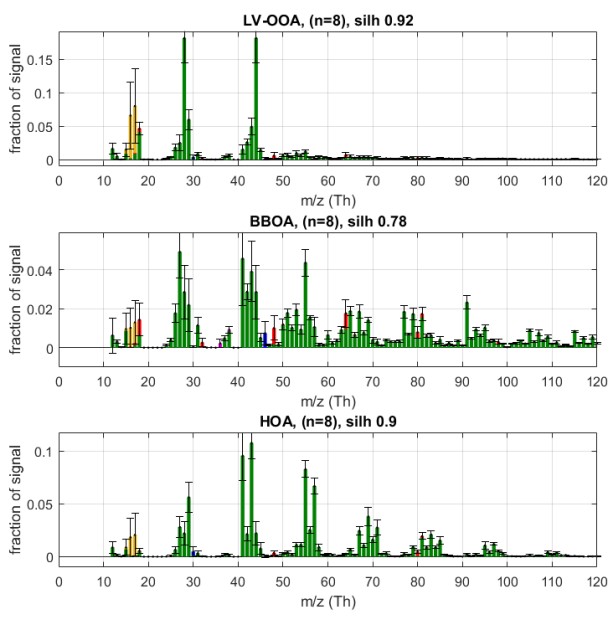

**Figure 4. Final silhouette-weighted reference spectra and variabilities for the r-CMB model components. (red: $SO_4$: blue $NO_3$; orange: $NH_4$; magenta: chlorides; green: org)**

## 3.2 Overview of r-CMB model results

### 3.2.1 Mass attribution and "default" AMS chemical speciation for r-CMB components

Tabulation of final explained variations (EV; Paatero, 2000; Canonaco et al., 2013) for the r-CMB model are shown in Table 2. The seven-component r-CMB model, explains $83 \pm 8$ % of the variation in loadings, when variation from low SNR variables is included, and $97 \pm 3$ %  when only residuals of variables with SNR > 2 are considered. The components with lowest loadings





(ON, HOA, AN) explain around 4 to 5 % of variation, which seems to roughly match the general rule of thumb, of PMF / ME2 being able to extract components of around 5 % of contribution (Ulbrich et al., 2009).

Model results for campaign VIII, especially regarding BBOA, are very different from other data sets, including the other cold season results available in e.g. set III. Upon closer examination, we attribute the VIII anomaly at least partly to pronounced

5    surface ionisation effects, discussed more in Section 3.4. While we consider the r-CMB results for campaign VIII too unreliable for use in models or further studies, we decided not to omit data set VIII, since other AMS data is likely also affected by the same processes, albeit to a lesser degree. The attribution of anomalies to exact processes is very difficult, and surface ionisation effects remain hard to quantify. We hope that reporting our results in full also furthers the discussion on surface ionisation in the AMS, and potentially helps other AMS users observing similar observations.

10   The composition of our r-CMB components is shown in Figure 5, panel (b), and the same in absolute in mass units in panel (a). The opposite visualisation, i.e. attribution of default species into r-CMB components, is similarly given for absolute mass concentration and relative units in Figure 5, panels (c) and (d). Unlike mass spectral variables and estimated EV, where signals at m/z are in units "nitrate equivalent mass" (RIE not applied), all mass concentrations reported are corrected for relative ionisation efficiency (see S.I., Sect S.4).

**Table 2. Explained variations (EV, in percent) for the r-CMB-like model.**

| Data set | | r-CMB component | | | | | | | Residual | | Rate of explanation | |
|---|---|---|---|---|---|---|---|---|---|---|---|---|
| nb | name | AS | LV | SV | BB | ON | HOA | AN | low SNR | high SNR | all | high SNR |
| I | May 2008 | 29 % | 15 % | 15 % | 3 % | 5 % | 5 % | 3 % | 21 % | 4 % | 75 % | 94 % |
| II | Sep 2008 | 34 % | 12 % | 14 % | 11 % | 5 % | 5 % | 3 % | 14 % | 3 % | 84 % | 97 % |
| III | Mar 2009 | 40 % | 19 % | 8 % | 8 % | 5 % | 2 % | 8 % | 9 % | 1 % | 90 % | 99 % |
| IV | May 2009 | 25 % | 16 % | 19 % | 14 % | 6 % | 5 % | 3 % | 10 % | 2 % | 88 % | 98 % |
| V | Jun 2009 | 18 % | 28 % | 23 % | 11 % | 4 % | 6 % | 2 % | 7 % | 1 % | 92 % | 99 % |
| VI | Aug 209 | 22 % | 26 % | 19 % | 9 % | 4 % | 5 % | 4 % | 10 % | 1 % | 89 % | 99 % |
| VII | Summer 2010 | 21 % | 25 % | 10 % | 7 % | 4 % | 4 % | 2 % | 21 % | 6 % | 73 % | 93 % |
| VIII | Winter 2010 | 25 % | 5 % | 1 % | 29 % | 7 % | 2 % | 4 % | 17 % | 9 % | 74 % | 89 % |
| mean | | 27 % | 18 % | 14 % | 12 % | 5 % | 4 % | 4 % | 14 % | 3 % | 83 % | 96 % |
| st.dev. | | 7 % | 8 % | 7 % | 8 % | 1 % | 1 % | 2 % | 5 % | 3 % | 8 % | 3 % |



**Figure 5.** "Default" chemical speciation for r-CMB components; mass loadings (upper left) and relative contributions (upper right) of default species in components. Apportionment of default species to r-CMB components by mass (lower left) and relative contribution (lower right).

Generally, the separation between the inorganics r-CMB components (AS, AN) and organics (LV-OOA, SV-OOA, BBOA, HOA) seems clear (Figure 5). Ammonium nitrate and sulphate components consist primarily of inorganic ion species (81 to 84 %), while for organic components the inorganic ion species contribution is small (LV-OOA: 8%, SV-OOA: 8%, BBOA: 6%, HOA: 3%). However, extensive oxidation of organics in aerosol typically results in formation of organic acids (Yatavelli et al., 2015; Vogel et al., 2013; Duplissy et al., 2011) and we hypothesise organic salt formation with [NH4+] could account for the notable 5 % mass contribution of ammonium to this aerosol type.

Explanations for the observed mixing of ion species can include (1) mixed emission profiles at sources, variabilities within a source type, as well as collocation of sources. (2) Atmospheric processes, such as mass transfer between the species by evaporation, condensation (e.g. Ye et al., 2016) as well as coagulation. (3) PMF / r-CMB modelling uncertainties. We will discuss the relative ratios and neutralisation balances of inorganic ion species in Section 3.3.2, in relation to inorganic salt



formation scheme. The interesting exception to the rather clear-cut ion species separation is the ON component, which contains 40 % of NO₃ species ions, and 41 % ions defined as organic. The possible interpretations for this distribution are further discussed in Section 3.3.2

As for the organics-inorganics division, the two speciations (default vs r-CMB) give similar results (Figure 6). For all the data sets combined, the default organic ion species ("org") explains an average 57 % of total aerosol mass at the site. Similarly, combining the mass of all organic-dominated components (LV-OOA, SV-OOA, BBOA, HOA and ON) results in 60 % mass fraction versus 40 % explained by ammonium nitrate (5 %) and ammonium sulphate (35 %) salts. The per-data set mass apportionment is presented in supplementary information (Figure S.9).

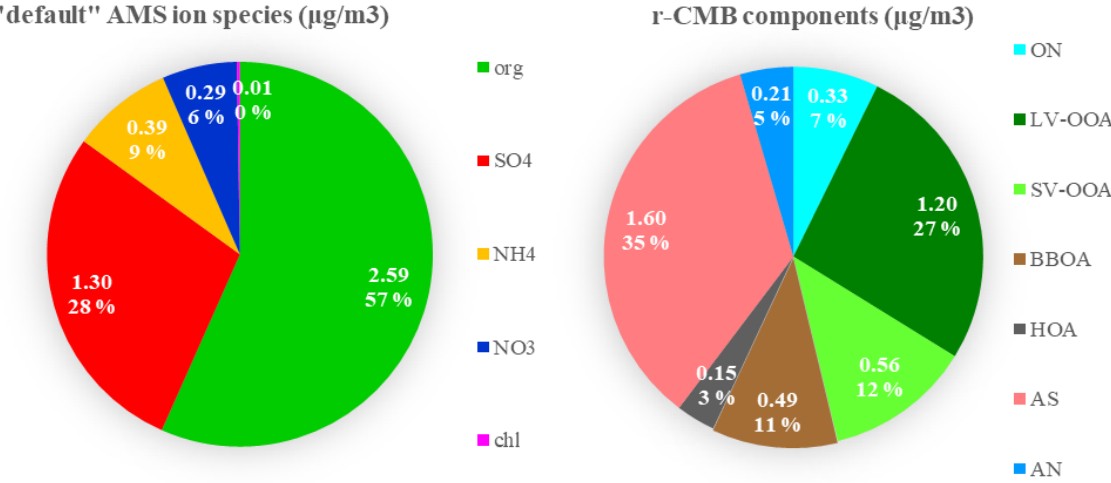

**Figure 6. Mass attribution in the default AMS speciation scheme (left) and by r-CMB components (right) for all the 8 data sets combined. Values are (data set length-weighted) averages for all data combined. Absolute mass concentrations are in units (μg/m³).**

**3.2.2 Spectral characteristics of organic components**

As discussed above, despite the mixing observed, the inorganic aerosol classes generally seem separate from organic aerosols. The scaled correlation values between inorganic and organic spectra are extremely low (S.I Sect S.8, Tables S.1 and S.2), indicating near-zero similarity and clear-cut separation between the inorganic and organic aerosol types by the clustering algorithm. For inter-correlations between the organics-dominated aerosol classes, the picture is somewhat more complex.

To understand the drivers for the separation of the organic aerosol types, we visualised the phase I (unconstrained PMF) and phase III (r-CMB) classification results with a projection of the clustering solutions onto a plane defined by an axis corresponding to estimated oxidation level and another connected to source type (P-III in Figue 7; P-I available in S.I, Fig. S.6.). Similar to Äijälä et al. (2017), we describe the oxidation level of the organic fraction of each component using the oxygen-to-carbon ratio (O:C) parametrisation of Aiken et al. (2008), and use the ratio between *f57:f57* fractions to imply source type. The O:C generally separates LV-OOA and SV-OOA species from each other and from the fresher aerosol classes. The



*f55:f57* ratio is typically used for differentiation between HOA and COA/BBOA (Mohr et al., 2012), but equally seems to set apart the biogenic SOA types from the anthropogenic aerosols (Äijälä et al., 2017). This is due to the low signal of *m/z* 57 Th, a typical anthropogenic spectral marker, originating from $C_4H_9^+$ and $C_3H_5O^+$ compounds (Mohr et al., 2012; Zhang et al., 2005).

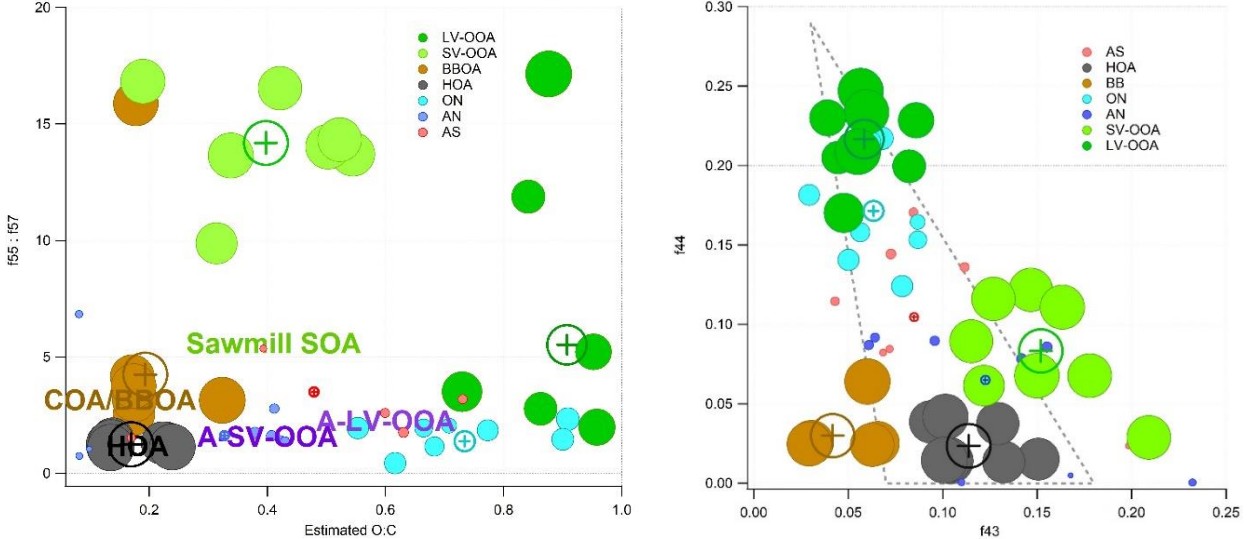

**Figure 7. (Left panel:) P-III (r-CMB) solution - cluster projections onto a *f55/f57* (Mohr et al., 2012), *O:C* (estimated, Aiken et al., 2008) plane. Circles correspond to the members of the cluster and the cross markers to cluster centroids. The text markers indicate respective positions of anthropogenic organic aerosol types from Äijälä et al. (2017). Marker size indicates organic mass fraction in spectra. Axes are truncated. (Right panel:) P-III solution, projected onto *f44, f43* plane (i.e. the 'Sally's triangle' plot; Ng et al., 2011). Circles correspond to objects in cluster and the cross markers to cluster centroids. Marker size indicates organic mass fraction in spectra. Dotted line marks the area where most laboratory data for organic aerosol falls (Ng et al., 2010).**

The LV-OOA aerosol type, characterised by the dominant *m/z* 44 and 28 Th signals is usually considered a highly oxidised aerosol type that results from oxidation of SV-OOA and various fresh emissions (among others Canonaco et al., 2015)). The *f55:f57* ratio of LV-OOA is considerably lower than for SV-OOA in both solutions, indicating inclusion of other sources beyond the *f57*-poor biogenic SOA contribution. SV-OOA, on the other hand, has the highest *f55:f57* ratio of the classes, hinting to the predominantly biogenic origin of the SV-OOA at the site. The difference is further amplified for phase II and III solutions compared to the unconstrained PMF. We hypothesise this change can result from improved differentiation between SV-OOA and the BBOA species (in P-II), as these aerosol types may be difficult to separate initially due to similar oxidation level and features of the spectra ($r_s^2 = 0.34$; Table S.3). The SV-OOA is characterised by the non-oxygen-containing ions at *m/z* 29, 43 and 55 Th (Mohr et al., 2009), as well as mass-to-charge *m/z* 53 Th signal ($C_4H_5^+$) typical of boreal forest biogenic background (e.g. Corrigan et al., 2013). The $NO_2^+/NO^+$ ratio of 0.10 for nitrate-containing SV-OOA reported by Hao et al (2014) matches our observations for the nitrates in SV-OOA ($NO_2^+/NO^+$ of $0.11 \pm 0.15$; Equations 7 and S.5). This may indicate presence of organonitrate species in the SV-OOA factor.



We also projected the P-I and P-III solutions to the (*f44, f43*) plane (P-III in Figure 7; P-I in S.I, Fig S.6), to produce a result comparable to the triangle plot by Ng et al. (2010). The result indicates a clear separation between the low and semi volatile aerosol types, as well as the primary combustion aerosols (HOA, BBOA), and the spectral shifts from phase 1 "bulk PMF" results to the final r-CMB model.

As stated in Section 3.1., the spectra of BBOA and HOA aerosol types match the previously published observations. The HOA spectrum is characterised by the ion series $C_nH_{2n+1}$ (*m/z* 29, 43, 57, 71, 85, 99 Th etc.) and $C_nH_{2n-1}$ (*m/z* 41, 55, 69, 83, 97 Th etc.) resulting from alkanes and aromatics from traffic emissions (diesel exhaust, lubricating oil; Chirico et al., 2010; Mohr et al., 2009; Canagaratna et al., 2004). The biomass burning organic aerosol levoglucosan marker signals at *m/z* 60 ($C_2H_4O_2^+$) and 73 Th ($C_3H_5O_2^+$) (Cubison et al., 2011; Schneider et al., 2006; Elsasser et al., 2012) are clearly identifiable in the BBOA

spectra (Figure 3; Figure 4), and set this class apart from HOA and SV-OOA with some similar features. The contribution of often biogenic signal at *m/z* 53 Th is also lower for BBOA than for the biogenic, semi-volatile SOA. The pronounced signal from aromatic ring (tropyllium cation $C_7H_7^+$) at *m/z* 91 Th is a typical result of fragmentation of aromatic hydrocarbon compounds (Lindon et al., 2016). As stated previously, we presume the BBOA class also encompasses any COA contributions, which are likely unresolvable as a separate class due to high spectral similarity (0.79; Sect 3.1.1).

In terms of spectral characteristics, the organic contributions of AS and AN classes fall somewhere between the distinct organic classes and offer little in terms of significant organic markers. Notably, the organics in the ON class exhibit some of the characteristics of LV-OOA and feature generally high *f44*. This may indicate high degree of oxidation of the organics for this aerosol type (Aiken et al., 2008). However, alternative plausible interpretations exist: AMS response from oxidation products of amine compounds and amine-nitrate salts feature similarly high f*44* (Murphy et al., 2007) as does a typical amine fragment

ion $C_2H_6N^+$ (McLafferty and Turecek, 1993). Furthermore, as discussed in Section 3.3.2, an equally plausible explanation would be inorganic nitrate salts such as $KNO_3$ contributing to this class in form of the Pieber et al (2016) thermal decomposition artefact. The contribution of *m/z* 55 and 57 Th signals to the ON species are both low and the ratio 1.37 of *f55:f57* is much lower than for the biogenic aerosol species. Without more detailed analysis, and due to the uncertainties surrounding the origins of this aerosol type (Section 3.3.2), it is difficult to say with any certainty if this is due to anthropogenic nature of this aerosol,

or e.g. due to fragmentation pattern of characteristic organic compounds in this aerosol type.

### 3.3 Comparisons with reference methods

### 3.3.1 Comparison with "traditional" ME-2 analysis for aerosol organic component

In order to evaluate the performance of the source apportionment approach presented in this study for organic aerosol, we compare our results to results only relying on the organic mass spectral fingerprints. Specifically, two data sets covered in this

study (data sets II and III; **Error! Reference source not found.**) were also included in the Crippa et al. (2014) analysis, which allows us to compared factorisation results directly. We chose to compare the Crippa et al. results to ours from data set II. We note that while there are minor differences in the pre-processing and corrections for data covered in Crippa et al (2014), the



factorisation input is very similar in both cases. The ME2 model used by Crippa and co-workers included only the organic spectra and apportioned its mass to four factors: LV-OOA, SV-OOA, BBOA and HOA. The latter two components were constrained using a HOA profile from an urban aerosol study in Paris (Crippa et al., 2013) and an average BBOA of those extracted for Mexico City, Mexico, and Houston, U.S (Ng et al., 2011). The allowed variability around these anchors for all

variables (*m/z*), were 5 % (HOA) and 30 % (BBOA).

We compared the solutions for Crippa et al. factorisation to our r-CMB model solution data set II, both for loadings (Figure 8) and profiles (Figure 9). Generally the solutions correlated highly – the loadings (F) and profiles (G) for LV-OOA (F: $r^2$=0.92; G: $r_s^2$= 0.96) and SV-OOA (*F*: 0.94; *G*: 0.99) agreed the closest, whilst the HOA also had high similarities (*F*: 0.85; *G*: 88). The BBOA factor / component correlated markedly less (*F*: 0.63; *G*: 0.42), which we hypothesise to be due to differences in

the anchors used, COA likely attributed to this class, high spectral similarity between SV-OOA and BBOA and the generally low loadings of BBOA observed at SMEAR II.

The discrepancy in distribution of absolute mass for the LV-OOA and SV-OOA components, indicated by the sub-unity slope, suggests the r-CMB model attributes a part of the organic mass from the SOA factors into BBOA, AS, AN and ON components, while HOA is represented rather identically in both models. A difference in  mass distribution between the results is to be

expected, considering the r-CMB model allows for organics in 7 components, while the model of Crippa et al. model only comprises 4 components. Generally, we take the similar results of the methods, as shown by the high correlation values, to indicate that inclusion of inorganics in the model does not significantly perturb modelling of the organics. We also note the r-CMB components included (HOA BBOA, LV-OOA, SV-OOA) are predominantly composed of organics (92 to 97 %; Figure 5), and the 4 components presented comprise 82 % of total organics.

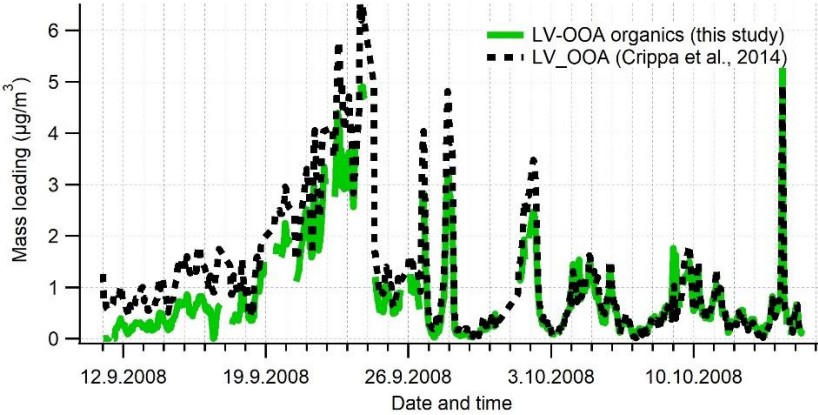
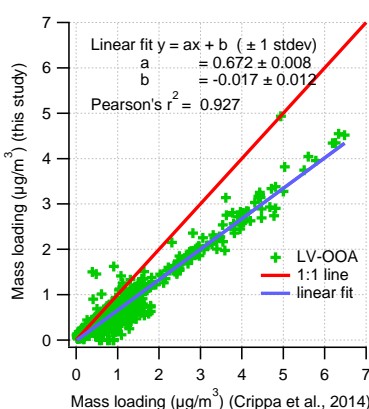









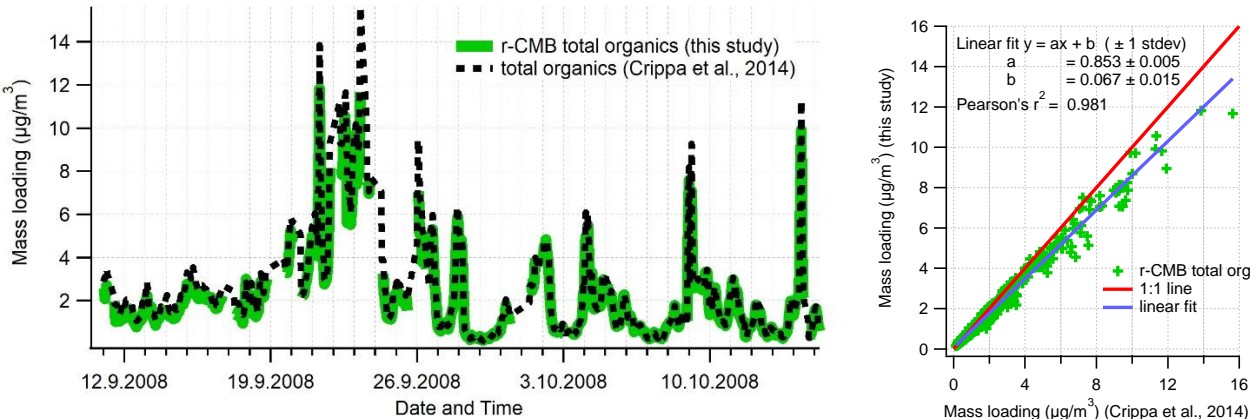

**Figure 8. Time series comparison of aerosol organic component with Crippa et al. (2014) for the September 2008 campaign (data set II). For comparability, only the organic part of r-CMB model components are considered. Data from this work has been averaged to 1 hr resolution. Organics in other r-CMB components (AS, AN, ON) are taken into account for the total amount but not shown separately. Discrepancy in total organics loading is due to differences in pre-processing values (e.g. ionisation efficiency, collection efficiency)**

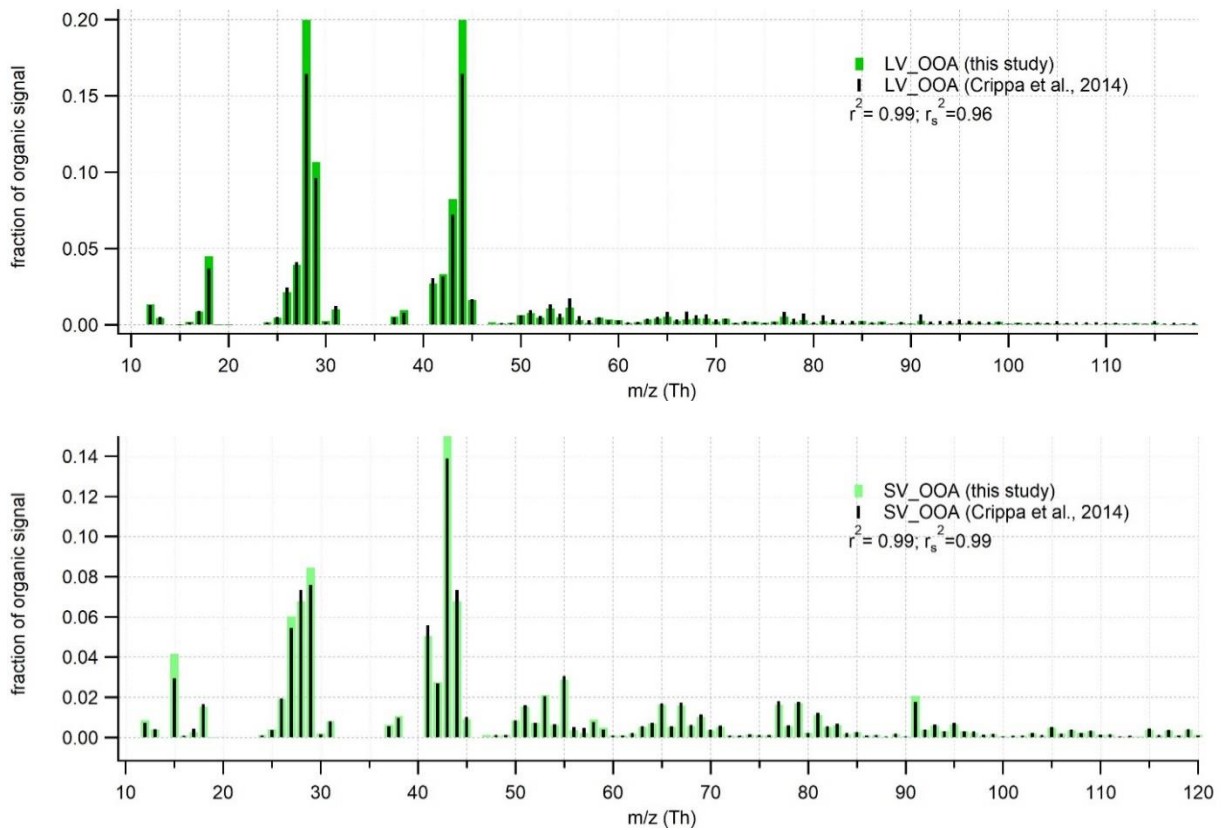





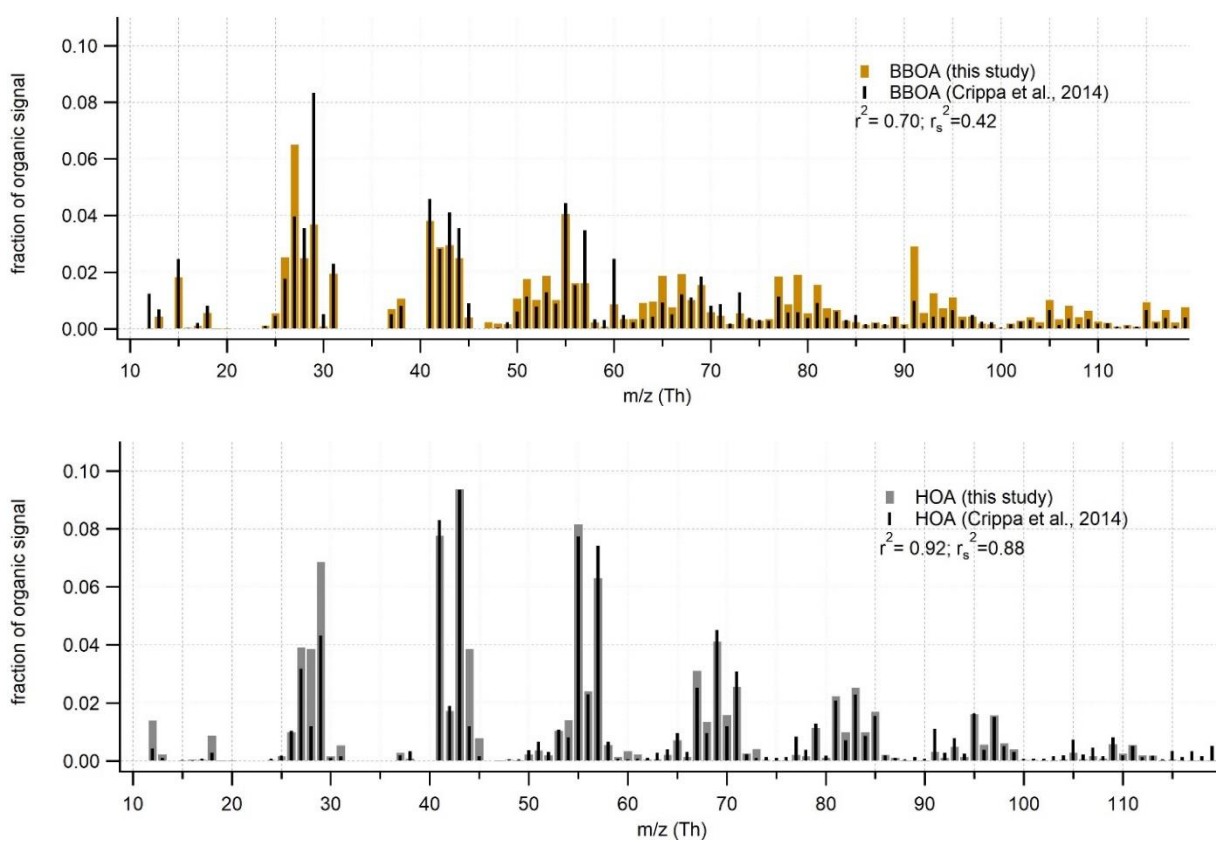

**Figure 9. Comparison of organic part of spectra with Crippa et al. (2014) for data set II. The r-CMB model results from this study are shown in colour, and the Crippa et al. spectra in black. For comparability, the Crippa et al. spectra were corrected for a difference in fragmentation tables used (included *m/z* 28 Th, updated to modern calculation of *m/z* 16, 17 and 18 Th organic signals) and total signal subsequently re-normalised to unity. Spectra similarity is evaluated using Pearson's squared correlation coefficients; unscaled ($r^2$) and with mass scaling ($r_s^2$).**

### 3.3.2 Comparison of inorganic salt and organic nitrogen results with reference methods

To evaluate the inorganic mass apportionment result, we compared the loadings from the r-CMB solution against the result from the inorganics apportionment scheme (Section 2.4.1). The comparison, again performed for data set II, is presented in Figure 10. We additionally compared the r-CMB ON component loadings with $orgNO_3$ mass estimate from the Kiendler-Scharr parameterisation (Equation 7; Section 2.4.2).

The loadings for the (r-CMB) AS component compare well with the combined $NH_4HSO_4$ + $(NH_4)_2SO_4$ + $H_2SO_4$ loading, indicating ammonium(bi)sulphate is described similarly by both models ($r^2$=0.92). We assume the r-CMB AS component to comprise of both $NH_4HSO_4$ and $(NH_4)_2SO_4$, which would very likely be classified together in PMF / clustering due to their high spectral similarity. For ammonium nitrate the correlation between loadings is very low ($r^2$ =0.16). Looking at the time series, the reason seems to be that the speciation scheme–based model often predicts total absence of AN, due to high amount of sulphate in aerosol. While the r-CMB model also generally estimates loadings to be low, they are clearly non-zero in the r-



CMB model. We take the result to reflect the assumption of complete and instantaneous, internal and external mixing of aerosol in the speciation scheme (Section 2.4.1).

The loading prediction for organic nitrogen by the speciation scheme model is similarly event-driven and the model results do not correlate. This is caused by the nitrate assignment to organonitrate class when not explained by $NH_4NO_3$. Same can be said for the excess $NH_4$ class, which corresponds to the $NH_4$ species in the other, mostly organic r-CMB factors, principally the LV-OOA; the ion balance scheme predicts zero concentration for many of the data points, an estimate not matching with the r-CMB-based result.

On these differences between the models, we note that the ion balance-based apportionment scheme is sensitive towards small changes in $NH_4$ concentrations, especially for data with generally low $NH_4$ concentrations, such as ours. A simple sensitivity estimate, available in supporting material (Sect. S.4) was performed for data set III. The result indicates that a 33 % change in $RIE_{NH4}$ changes the component mass concentrations on average 5% for AS; 56 % for AN, 66 % for orgNO3 and 164 % for excess_NH4 components. On the other hand, the r-CMB model is rather insensitive to error in RIE estimate, since (1) the spectra in factorisation and clustering have the variables' signals in "$NO_3$ equivalent mass concentration" units, which is not (yet) corrected for RIE of different species, (2) mass scaling causes low mass signals such as $NH_4$ fragments (*m/z* 15 to 17 Th) to weight less (relative to higher *m/z* variables) for determining the solution, and (3) $NH_4$ seem not to be an unique marker of any of the classes. We therefore suggest a factorisation-based model such as the r-CMB model presented here is much more robust for resolving speciation of inorganic aerosol components. The sensitivity test (S.I, Sect S.4) also indicates that the temporal differences between the ion balance scheme and r-CMB are not explained by a difference in $RIE_{NH4}$. Thus, the reasons for the discrepancies are more likely related to the unrealistic assumptions of the inorganics apportionment model.








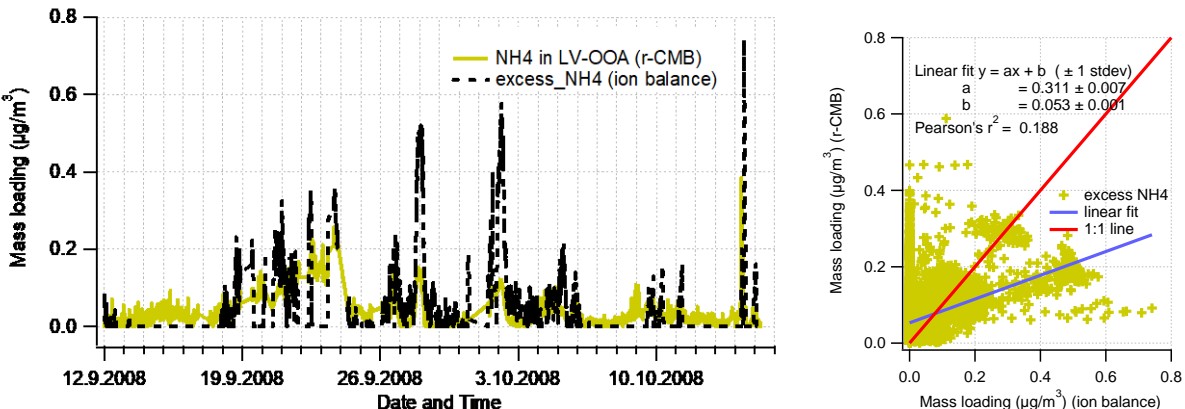

**Figure 10. Comparison of Inorganics apportionment methods (r-CMB and ion balance scheme. The estimates from the ion balance scheme (Section 2.4.1) are shown in black, and the r-CMB model results in colour. The linear fits (right panels) represent the data poorly due to high amount of zero-value points and outliers.**

In addition to deriving organic nitrogen mass from the ion balance scheme, we compared the r-CMB-derived ON loading with the Kiendler-Scharr method for estimating the orgNO3 mass loading (Equation 6). The comparison, shown in Figure 11, indicates that the two methods produce a very similar result for organic nitrogen mass ($r^2 = 0.94$). The discrepancy in absolute mass is likely explained by the difference in the ratio values (R) used for Equation 6 parameterisation, and those featured in the r-CMB AN and ON components ($R_{AN} = 0.56$; $R_{ON} = 0.05$).

The similarity to Kiendler-Scharr parameterisation result does seem to support the interpretation of nitrogen component in ON as organonitrate (orgNO3). Some similarities in temporal behaviour between the ON component and (non-quantitattive) $K^+$ ions were observed, potentially suggesting thermal ionisation of Potassium salts (e.g. $KNO^3$) might contribute an unknown fraction to ON (S.I, Sect S.11). Also, 63% of chloride ions species associate with the ON component. The reason is unclear, and although chloride signals were very low in general, we cannot rule out that some of the ON component could still be explained by other chemical compositions than organonitrate.

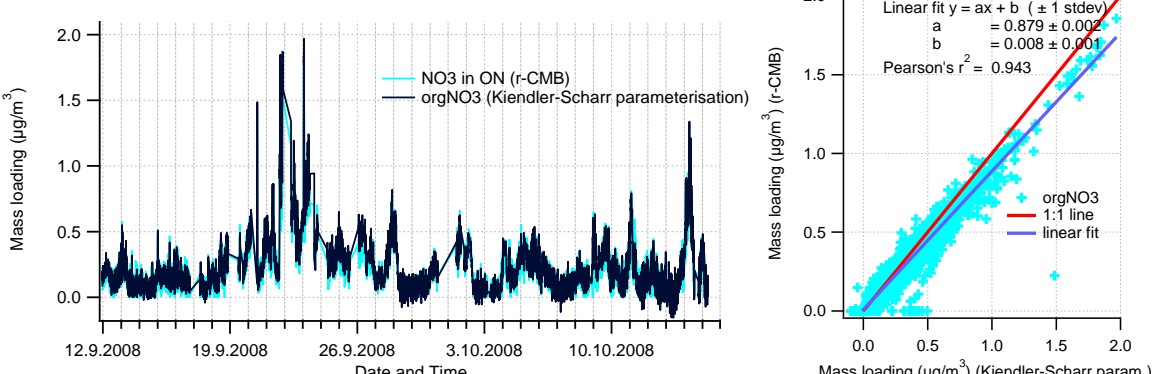

**Figure 11. Comparison of Kiendler-Scharr parametrisation (Kiendler-Scharr et al., 2016; black line; moving median filter for 11 points window applied; $R_{calib} = 0.42$, $R_{orgNO3} = 0.1$) for organonitrate with NO₃ ion species in ON factor from our r-CMB model (in colour).**



The $NO_3$ : org ratio of our ON factor is close to unity (Figure 5), while e.g. Farmer et al. (2010) report a nitrogen-to-carbon ratio of 0.04, and oxygen-to-carbon of 0.25 for AMS spectra of organonitrate standards. However, several factors are likely to affect the $NO_3$ : org ratio observable in atmospheric ON factorisations. Firstly, two different pathways for organonitrates exist: i) the primarily daytime reactions of organic peroxy radicals with NO (Orlando and Tyndall, 2012), and ii) the $NO_3$-radical initiated oxidation of unsaturated compounds during night-time (Peräkylä et al., 2014). While the nitrate functionality in all these reactions are identical, the organic part can be vastly different, as peroxy radicals are formed in almost all atmospheric oxidation reactions, irrespective of oxidant (e.g. OH or ozone) or VOC (biogenic or anthropogenic). Therefore, it is not to be expected that a specific organic spectrum should be linked to the organic nitrate functionality. Secondly, as described by e.g. Lee et al. (2016), the particle phase lifetime of organonitrates is of the order of hours with respect to hydrolysis. This reaction will convert the nitrate functionality to nitric acid, while the organic part remains intact, except for the conversion of the -$ONO_2$ group to -OH. This conversion will only have a small impact on the volatility of the organic molecule (e.g. Kroll and Seinfeld, 2008), while the nitric acid may well evaporate in the fairly low-ammonia boreal forest environment. Taken together, the diverse formation pathways as well as the atmospheric processing are likely to cause ON spectra retrieved from ambient air factorisations to look different from e.g. freshly formed organic aerosol from organonitrate standards, such as those used by Farmer et al. (2010). We therefore avoid putting too much emphasis on the organic parts observed in our ON factor.

### 3.4 Outlier observations

During the course of our analysis we encountered some anomalous observations likely stemming from surface ionisation effects, *i.e.* molecules being thermally ionised at the heater surface rather than at the ionisation region by electron impact. A thorough review and discussion on AMS related surface ionisation effects was recently published by Drewnick and colleagues (2015). Drewnick et al. emphasise that the division between refractory and non-refractory aerosol is not binary, and there exist a number of semi-refractory compounds, that the AMS can measure, albeit non-quantitatively.

Our observations on extracted "outlier" PMF factors from the different phases of analysis match well with the finding and calculations of Drewnick et al (2015), as well as other similar AMS observations published. In Figure 12, we present the outlier clusters from phase I classification solution that were excluded from further analysis due to low number of occurrences or/and questionable interpretability. The emergence of most of these spectra are likely attributable to over-resolution or questionable separation of main PMF factors, due to setting the number of PMF factors to 10. Despite their questionable value for the main analysis, we find they contain many potentially interesting mass spectral features, and seem not to emerge by chance. Below we will present some hypotheses on their possible interpretation.





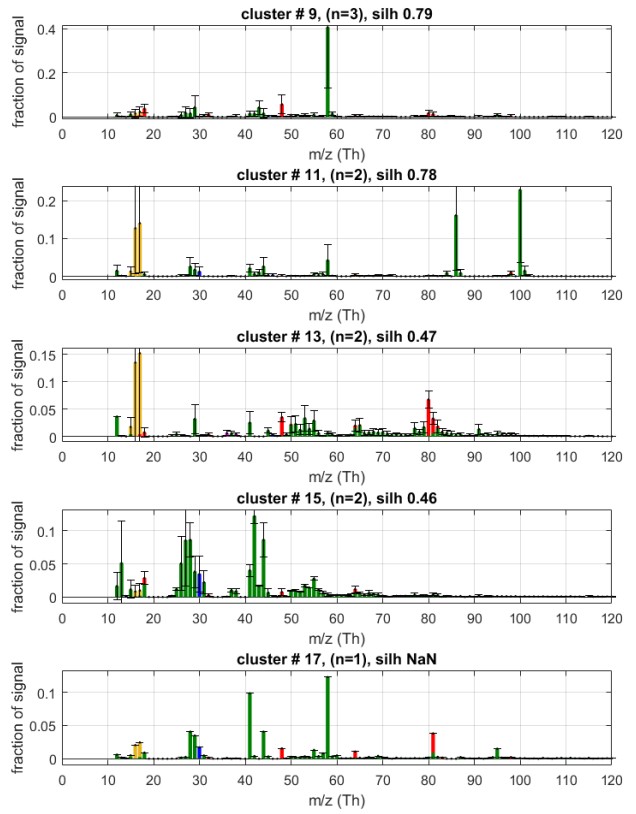

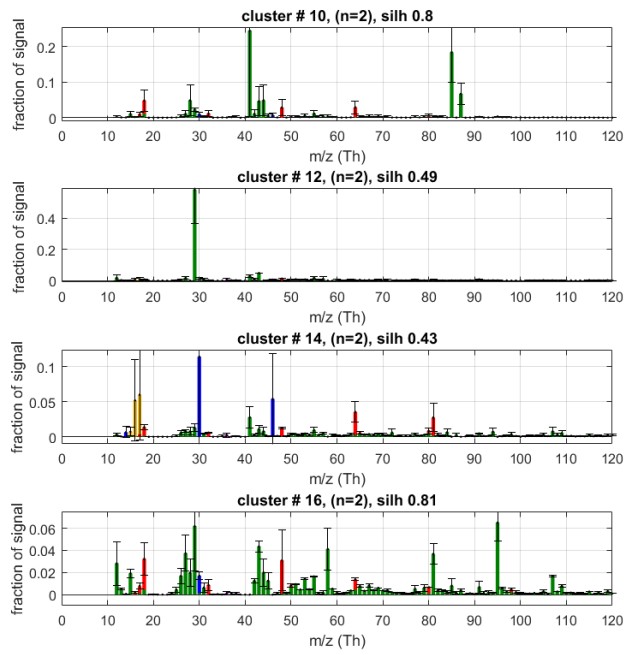

**Figure 12. Spectra of outlier clusters (#9 to #17) for P-I. The spectra for these outlier classes were omitted from our analysis due to not meeting the criteria of (1) occurrence or/and (2) interpretability (on an acceptable level). Despite their mostly speculative value, many of them feature some chemically interesting characteristics, potentially pointing to presence of amines (signals at *m/z* 58, 86 and 100 Th; clusters #9, #11 and #17), alkali metals (<sup>85</sup>Rb, <sup>87</sup>Rb; #10), cycloalkanes (signals at series *m/z* 69, 79, 81, 95, 107 and 109 Th; #16), and organic sulphate (signal at *m/z* 80, 81 Th; #13, #17), as well as effects of surface ionisation (<sup>41</sup>K<sup>+</sup>; <sup>39</sup>K<sup>+++</sup>; #10, #17) and a likely artefact from poor airbeam correction (signal at *m/z* 29 Th; #12). (red: SO₄: blue NO₃; orange: NH₄; magenta: chlorides; green: org)**

### 3.4.1 Surface ionisation and data correction artefacts

Drewnick et al. (2015) note that the main semi-refractory elements eligible for ionisation in the AMS are Cd (*m/z* 112 Th), Cs (132 Th), Hg (200 Th), K (39 Th), Na (23 Th), Rb (85 Th) and Se (79 Th). The proneness of potassium (K) and sodium (Na) for non-quantitative thermal ionisation effects in the AMS is well known (e.g. Allan et al., 2003a), which is also why they are excluded from AMS (quantitative) data analysis. Although the main potassium isotope omitted, the $^{41}$K isotope (with 6,7 % relative abundance; Haynes, 2014), is not, and a correction applied in fragmentation table instead. The K-derived signals were especially prominent in data set VIII (see S.I Fig S.7), with contributions of 1 to 2 order of magnitudes higher than highest well-behaving signals such as *m/z* 44 Th or 48 Th. We hypothesise the strong signals at *m/z* 41 Th observable in many of the outlier spectra (cluster #10, #15, and #17) may be due to insufficient accuracy of the $^{41}$K isotope correction.

A similar data processing/correction artefact is likely seen in cluster #12 with a lone, dominant signal at *m/z* 29 Th. This mass-to-charge is a problematic one for lower-resolution AMS data due to contribution of $^{29}$N$_2$ isotopic peak, and location on the



slope of the enormous $N_2$ peak at $m/z$ 28 Th. Although the signal at $m/z$ 29 Th is corrected for the (measured) isotope contribution, even a slight mismatch in the correction results in notable error in the estimation of the organic signal fraction at $m/z$ 29 Th. We attribute this problem specifically to the scarce availability of filters for the earliest sets of data.

### 3.4.2 Alkali metals

The prominent signals at $m/z$ 85 and 87 Th for cluster #10 correspond to Rubidium alkali metal ions, and their respective ratios ($m/z$ 85 Th signal : $m/z$ 87 Th signal = 73.2 : 26.8 ) to what we would expect based on isotopic distribution of Rb observed in nature ($^{85}$Rb : $^{87}$Rb 72.2 % : 27.8 %; Haynes, 2014)). Unlike for the potassium signal, the temporal behaviour of the factors corresponding to cluster #10 is highly plume-like. Preliminary analysis of wind direction shows the plume direction to correspond to the arrival direction from the district heating plant (co-located with a sawmill and a pellet factory) at Juupajoki, 5 km due south-east (Supplementary information, Sect S.12). Similar observations of Rubidium from coal burning are previously published by Irei et al. (2014). It seems likely that this aerosol class would originate from the heating plant.

### 3.4.3 Organic nitrogen and sulphur

As for the signals often attributed to amines at 86 and 100 Th, (Mclafferty, 1959), featured in cluster #11, in absence of alternative explanation for the 100 and 86 signals, we are inclined to believe they actually represent atmospheric amines. The cluster spectrum corresponds also to the spectra of pollution plumes, extracted for data sets I to III in our previous study on pollution events (Äijälä et al., 2017). We note amines are also reported to be prone to surface ionisation, and e.g. trimethylamine is thermally ionised above temperatures 300°C, with high thermal ionisation efficiency at 600°C (50% of the maximum efficiency observed at around 350°C; Rasulev and Zandberg, 1988)). It thus seems plausible surface ionisation effects could contribute to the amine observations as well. In our earlier work (Äijälä et al., 2017), we also attributed similar spectral signal at $m/z$ 58 Th to amines ($C_3H_8N^+$). However, in light of the recent results of Drewnick et al (2015) on surface ionisation of NaCl, and the detachment of $m/z$ 58 Th signal from the class of other amine-attributed signals at 86 and 100 Th, another plausible explanation for the $m/z$ 58 Th signal observed in clusters #9, #11, #16 and #17 exists. Namely, we find it plausible such a spectrum would arise from surface ionisation of sodium chloride and thus represent atmospheric $NaCl^+$.

Clusters #13, #15 and 16 are interesting from the viewpoint of organonitrates and sulphates. Nitrate signal in clusters #15 and #16 is composed mostly of $m/z$ 30 Th signal, with negligible $m/z$ 46 Th contribution. With the high organic contribution, this would make these classes potential candidates for containing organonitrates. However, an equally plausible explanation is the surface ionisation of $KNO_3$, discussed previously. The pronounced signals at $m/z$ 80 and/or 81 Th featured in cluster #13, #14 and #17 are likely explained by humidity-induced fragmentation changes in ionisation of sulphate species, (particularly $H_2SO_4$ and $SO_3$; Drewnick et al., 2015). We do note organosulphur-containing samples characterised by Farmer at al. (2010) also feature increased ratio of $m/z$ 80 and 81 Th signals compared to $(NH_4)_2SO_4$, so we can not rule out organic sulphate contribution.



### 3.4.4 Cycloalkanes

Finally, we wish to draw attention to the ion series of cluster #16, with prominent organic signals at (69), 79, 81, 95, 107 and 109 Th, which have been connected to cycloalkanes (McLafferty and Turecek, 1993; Alfarra et al., 2004). Cycloalkanes are common in e.g. lubricating oils (Liang et al., 2018), which are an important, even dominant, component in traffic emissions (Worton et al., 2014). The closest literature match on ambient observations we found was the study of Takami and colleagues (2007), where they observed similar high concentrations of mass-to-charge 95, 107, 109 Th, as well as 58 and 85 Th, but were unable to attribute the observation to a specific source.

### 4. Conclusions

We performed a synthesis analysis on eight AMS data sets from a boreal forest site, and constructed a data-driven chemical mass balance type of receptor model, with relaxed constraints on the component profiles (r-CMB). Notably, the data comprised both inorganic and organic aerosol components. The resulting 7-component model explained $83 \pm 8$ % of variability in data ($96 \pm 3$ % with low-SNR variables excluded). The model components for the SMEAR II boreal forest site were, in order of average aerosol mass contribution: Ammonium sulphate (35 % mass fraction), LV-OOA (27 %), SV-OOA (12 %), BBOA (11 %) Organic nitrogen (7 %) Ammonium nitrate (5 %) and HOA (3 %).

Remarkably, organic nitrogen seems a larger component than ammonium nitrate for the site. However, ambiguity remains on the interpretation of the organic nitrogen class as organonitrate, prompting caution against casual use of $NO_2^+:NO^+$ fragmentation ratio as a sole organonitrate proxy. COA was not resolved separately, presumably due to high spectral similarity with BBOA and low mass contribution to SMEAR II aerosol, and is most likely included to the BBOA component. Other, minor aerosol groups that were not included in the model feature characteristics potentially indicative of amine-dominated aerosols, coal combustion aerosol with alkali metals (Rubidium, Cesium), as well as hints of cycloalkanes and organosulphates. We presume many of these observations may arise from by surface ionisation processes, and as such they may not be currently quantifiable in mass. Their corroboration, quantification, and connection to emission sources or thermal ionisation effects require for further study.

We suggest inorganics should be routinely included in factorisation of AMS data due to the high demand of such data in aerosol models. We wish specifically to point out that adding the inorganic information is easy and only requires applying the same tried-and-tested data processing and uses the same error model as for organics. While inclusion of inorganics does diminish the relative weight organics carry in the analysis, and thus may hinder extraction organic factors comprising very low fraction (<5 %) of total mass (Ulbrich et al., 2009), we argue that the added information value of inorganic speciation makes up for this. Compared to organics only analyses, inclusion of inorganic data increases direct usability of AMS data for physicochemical aerosol models. We also demonstrate factorisation-based speciation provide a more realistic and robust, less assumption-dependant and calibration-sensitive, speciation than simplistic ion balance schemes.



The classification methods presented here for evaluating factor analysis output can also have direct use in applications that produce large quantities of discrete aerosol spectral data, such as deriving factorisation error estimates via bootstrapping analysis (Osborne et al., 2014; Brown et al., 2015). With further development, we find it likely a two-step analysis (exploratory factorisation + classification → r-CMB) would be a viable option for increasingly unsupervised and less analyst biased AMS

data analysis.

We would also encourage further development of combined statistical methods for improved mass spectral feature extraction and parametrisations for mass spectra, as they will enable future machine learning applications for data analysis. Drawing from the comprehensive information available in current size-resolved aerosol mass spectrometric data, it seems likely that advanced machine learning methods (such as data reduction combined with predictive neural networking (e.g. Burns and Whitesides,

1993; Gasteiger and Zupan, 1993) will likely provide new, improved ways to model aerosol physicochemical properties like hygroscopicity, volatility and optics in the near future.

## Data availability

Data used in this study is available from the contact author. r-CMB component profiles will be made available in the AMS spectral database ([http://cires1.colorado.edu/jimenez-group/AMSsd/](http://cires1.colorado.edu/jimenez-group/AMSsd/)) upon publication.

## Author contribution

*Conceptualisation:* MÄ, ME formulated the study. *Investigation and data curation*: MÄ, HJ, ME collected and curated the experimental data. PA*, ES*, JL*, HL* TP* provided technical support for the experiments. *Formal analysis, methodology, visualisation*: MÄ, supported by KD, designed and performed the statistical analysis and data visualisations. *Validation:* MÄ and KD reviewed the data quality and reproducibility. *Software, methodology*: FC designed and supported the SoFi analysis

software. DS*, LW* provided support for Squirrel software. *Writing:* MÄ wrote the original draft, which was reviewed, commented and edited by all the authors. *Funding acquisition, resources, project administration, supervision*: ME, MK, AP, TP and DW* supported and supervised the research.

Contributor roles in *italic* correspond to the taxonomy of CASRAI's CRediT definitions ([https://casrai.org/credit/](https://casrai.org/credit/))

* see Acknowledgements

## Competing interests

The authors declare they have no conflict of interest.



**Acknowledgements**

We wish to thank the technical staff at INAR and SMEAR II (Pasi Aalto, Erkki Siivola, Heikki Laakso, Toivo Pohja, Veijo Hiltunen and Janne Levula) for valuable support during the years 2008-2010, in acquiring the data sets analysed here. We thank Prof. Douglas Worsnop for pioneering work in starting the AMS studies at University of Helsinki, and the valuable

5  insightful discussions on AMS data analysis and interpretation. We also gratefully acknowledge the friendly support staff at Aerodyne Research (especially Donna Sueper and Leah Williams) for their help on data analytical questions.

The research was supported by the following programs: the European Commission FP6 projects EUCAARI (036833-2), FP7 ACTRIS (262254), the Horizon 2020 project ACTRIS-2 (654109), ERC Grant COALA (638703), the Finnish COE project CRAICC (272041) and the Academy of Finland COE in Atmospheric Science (2008–2019).

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
