# Peer review of "Constructing a data-driven receptor model for organic and inorganic aerosol - a synthesis analysis of eight mass spectrometric data sets from a boreal forest site"

_Atmospheric Chemistry and Physics, 2018_

## Referee Comment (RC1) · Anonymous Referee #1 · 17 Nov 2018

Review for Constructing a data-driven receptor model for organic and inorganic aerosol
- a synthesis analysis of eight mass spectrometric data sets from a boreal forest site
by Mikko Äijälä et al.

This paper describes the development of a new way to perform source apportionment,
analysing eight different mass spectrometric datasets. The topic of this paper is in-
teresting to the community and will help on improving future source apportionment
studies. I recommend this paper for publication after the authors address the following
comments.

[Figure]

Specific comments.

Introduction. The factorization tools used in this study are PMF and ME-2. However, the authors do not mention ME-2 in the introduction. It would be good to read how ME-2 helps on separating profiles when PMF struggles to do so.

Page 3 line 25. I think the authors want to stress the importance of local anthropogenic sources in the last paragraph. If that is the case, rephrase the last paragraph for something like: While previous studies have found biogenic SOA and long-range transport from industrial regions to be important, local anthropogenic aerosol sources are also present. At the moment that paragraph is confusing, please rephrase it.

Section 2.3.1. When describing ME2, the method used to constrain solutions should be explained as well. Page 16 Second paragraph. When talking about BBOA and COA, one of the main differences between these factors is the diurnal profile, COA usually shows a small peak at lunch time and then increases in the evening. Do the authors had a look at diurnal profiles to differentiate between COA and BBOA? Diurnal profiles provide interesting information about the different profiles identified.

Technical corrections

A number of typos were found in the manuscript. I suggest to go through the document again and correct the typos. These are a few minor comments I would like to provide.

Page 2 line 2. Change effects for properties Page 2 line 3. Change almost for near. Page 3 line 2. Provide the references to the previous literature. Page 4 line 2. Delete the word "to" before 2008. Page 4 table 1. Perhaps add a column with the number of months for an easier comparison. Page 4 line 10. Please define if it was a compact or a high resolution AMS. Page 6 line 30. Provide references where ME-2 has been used to partially constrain solutions. Page 9 line 4. Change: 'There exist' for 'There are' Page 21. "F57:f57 fractions", it should be f55:f57.

---

## Referee Comment (RC2) · Anonymous Referee #2 · 4 Dec 2018

The manuscript "Constructing a data-driven receptor model for organic and inorganic aerosol - a synthesis analysis of eight mass spectrometric data sets from a boreal forest site" introduces a novel receptor model for organic and inorganic aerosol measured in Hyytiälä, Finland between 2008 and 2010. The measurements were performed with a CToF aerosol mass spectrometer and receptor model was applied to unit-mass-resolution mass spectra. The benefit of this receptor model for organic and inorganic aerosol over traditional PMF/ME2 for organic MS is that it yields useful information for the modelling of submicron atmospheric aerosols physical and chemical properties,

and the results illuminates the division between organic and inorganic aerosol types and dynamics of salt formation in aerosol.

General comments

This paper is written precisely, logically and clearly. It presents novel methodology and scientific results that are very useful for atmospheric scientists, both experimentalists and modelers. I think this paper should be published in ACP after minor revision.

Specific comments

1. Page 1, Abstract; line 24; "simplistic inorganics apportionment methods" is unclear to me, do you mean PMF/ME2?

2. Page 4, CToF-AMS; what is the mass resolving power of CToF? Is it possible to use high resolution mass spectra instead of UMR-MS? How reliably you can identify alkalimetals, especially rubidium, with CToF?

3. Page 5, Section 2.2.2 Data preparation and down-weighting; Could you explain here (shortly) how RIEs and fragmentation table were taken into account?

4. Page 6, line 28-29, "all the source profiles are constrained, but allowed to vary within narrow limits", what alpha-value (constraning value)?

5. Page 11, line 1; "we applied an ion ratio Rcalib = 0.42, taken as the average of mass spectrum based AN calibrations (S.I Sect S.6)". Do you mean m/z 46/30 in Fig. S3? It seems to be much larger than 0.42. In general, Figure S3 is very difficult to read because it is unclear and has very small fonts. Please improve the quality of the figure.

6. Page 15, line 16-18; similar comment, for cluster #8 m/z 46/30=0.44, according to Figure 3 the ratio is larger

7. Page 30, line 12, what is the origin of KNO3 in Hyytiälä?

8. Results and discussion in general: diurnal trends of the factors are not utilized at

all when interpreting and identifying aerosol components, and similarly, any auxiliary gas (or particle) data is not exploited. Couldn't this additional data help interpreting the results?

9. Supplemental material, Figure S5; could you add total mass for each data set?

Technical corrections:

10. Page 11, line 18-19, second parenthesis is missing

11. Page 30, line 11; non-quantitative
* * *

---

## Author Comment (AC1) · 21 Dec 2018

**Anonymous Referee #1** (authors' responses in blue)

Review for Constructing a data-driven receptor model for organic and inorganic aerosol
- a synthesis analysis of eight mass spectrometric data sets from a boreal forest site
by Mikko Äijälä et al.

This paper describes the development of a new way to perform source apportionment, analysing eight different mass spectrometric datasets. The topic of this paper is interesting to the community and will help on improving future source apportionment studies. I recommend this paper for publication after the authors address the following comments.

We thank the Anonymous Referee for his/her time in reviewing the manuscript and appreciate the constructive comments.

Specific comments.

Introduction. The factorization tools used in this study are PMF and ME-2. However, the authors do not mention ME-2 in the introduction. It would be good to read how ME-2 helps on separating profiles when PMF struggles to do so.

To keep the introduction short and present an introductiory storyline for the reader, we have not listed all the statistical algorithms in this section (kmeans, ME-2, similarity metrics, etc.), but approach the topic from a broader perspective of why a chemometric approach is needed in this field. To accommodate the Referee's valid point that constraining is sometimes needed, we added a mention of this to the discussion of analyst subjective choices: *"Specifically, while analyst imposed additional constraints in factorisation may sometimes be required to reduce rotational uncertainty and extract minor factors in data (e.g. Canonaco et al., 2013; Crippa et al. 2014) such procedures are especially prone to analyst subjective decisions."*

Page 3 line 25. I think the authors want to stress the importance of local anthropogenic sources in the last paragraph. If that is the case, rephrase the last paragraph for something like: While previous studies have found biogenic SOA and long-range transport from industrial regions to be important, local anthropogenic aerosol sources are also present. At the moment that paragraph is confusing, please rephrase it.

We rephrased and re-ordered the paragraph. Here we wanted to list the known anthropogenic aerosol sources, so we re-arranged them by distance (long-range -> regional -> local), to clarify the paragraph.

Section 2.3.1. When describing ME2, the method used to constrain solutions should be explained as well.

This is a valid point, which was probably not clarified enough. We now added a short explanation: *"In this study, when ME-2 constraints were applied to the factor profiles, we set upper and lower bounds for the allowed profile solutions. The bounds were based on variability estimates obtained from earlier analysis, as explained later, in Sect. 2.5."* We also added an explanation of the ME-2

limits in 2.5: *"The allowed variabilities were constrained by setting upper and lower bounds (the estimated variability ranges from the previous phase) for factor profiles."* (See also response to Referee 2, comment #4).

Page 16 Second paragraph. When talking about BBOA and COA, one of the main differences between these factors is the diurnal profile, COA usually shows a small peak at lunch time and then increases in the evening. Do the authors had a look at diurnal profiles to differentiate between COA and BBOA? Diurnal profiles provide interesting information about the different profiles identified.

We added the diurnal analysis to the supplementary material to include this information. However, regarding the BBOA and COA, the temporal behaviour stated above only applies for urban environments or close-by sources. As described in Sect. 2.1.1, most anthropogenic aerosol (besides the plumes from the station itself, which is insignificant in terms of observed aerosol mass) is transported from 5 to 50 kilometres away. With common wind speeds of a few m/s (or 10-15 km/h) the distribution already becomes somewhat smeared, so we do not expect to see e.g. clear lunch-time peaks. We added a paragraph to 3.1.3: *"Diurnal cycles of the components for the entirety of data are available in S.I (Figure S.12). Due to the rural setting of the site and the generally long transport times of aerosol before reaching the site, diurnal cycles for the various aerosol types are not as characteristic as they would be for urban measurements (for e.g. temporal trends of HOA and BBOA). Also due to seasonal differences, the variability between data sets is considerable, resulting in high uncertainty in interpretation. The daily cycles are likely a mixed product of source emissions, boundary layer dynamics and aerosol temperature response. While of interest, disentangling these processes is beyond the topic of this study."*

[Figure]

Technical corrections

A number of typos were found in the manuscript. I suggest to go through the document again and correct the typos. These are a few minor comments I would like to provide.

Page 2 line 2. Change effects for properties
Changed

Page 2 line 3. Change almost for near.
Changed (p.2,l.5)

Page 3 line 2. Provide the references to the previous literature.
This is just an opening remark that we will not cover all the technical details in this article. The references are included in the relevant sections. We added:
"[Our instrumentation, data processing, measurement site and analysis algorithms have been conscientiously described in previous literature,] to which we refer in the corresponding sections."

Page 4 line 2. Delete the word "to" before 2008. Page 4 table 1. Perhaps add a column with the number of months for an easier comparison.
Corrected. Good suggestion. We added a graphical table indicating monthly data availability to Sect. 2.1.2.

| | Jan | Feb | Mar | Apr | May | Jun | Jul | Aug | Sep | Oct | Nov | Dec |
|---|---|---|---|---|---|---|---|---|---|---|---|---|
| **2008** | - | - | - | - | 65 % | 20 % | - | - | 70 % | 48 % | - | - |
| **2009** | - | - | 94 % | 23 % | 90 % | 63 % | 81 % | 87 % | 63 % | - | - | - |
| **2010** | - | - | - | - | - | - | 74 % | 68 % | - | - | 47 % | 100 % |
| **2011** | 23 % | - | - | - | - | - | - | - | - | - | - | - |

Page 4 line 10. Please define if it was a compact or a high resolution AMS.
This information is already in the next sentence: "AMS instruments in general have been described by Canagaratna et al. (2007), and the compact ToF analyser version (CToF) used in this study by Drewnick et al. (2005)" (p.4.,l.12)

Page 6 line 30. Provide references where ME-2 has been used to partially constrain solutions.
We added here: *"[…] allowed to vary within narrow limits (derived from variability estimates; see Sect 2.5). Variability estimate of the final model is available in S.I (Figure S.13)."*

Page 9 line 4. Change: 'There exist' for 'There are'
We do not see any difference in meaning or clarity here, but rephrased the sentence for fluency: *"A variety of aerosol inorganic equilibrium models exist, and are typically used as modules…"*

Page 21. "F57:f57 fractions", it should be f55:f57.
Corrected.

**Anonymous Referee #2** (authors' responses in blue)

The manuscript "Constructing a data-driven receptor model for organic and inorganic aerosol - a synthesis analysis of eight mass spectrometric data sets from a boreal forest site" introduces a novel receptor model for organic and inorganic aerosol measured in Hyytiälä, Finland between 2008 and 2010. The measurements were performed with a CToF aerosol mass spectrometer and receptor model was applied to unit-mass resolution mass spectra. The benefit of this receptor model for organic and inorganic aerosol over traditional PMF/ME2 for organic MS is that it yields useful information for the modelling of submicron atmospheric aerosols physical and chemical properties, and the results illuminates the division between organic and inorganic aerosol types and dynamics of salt formation in aerosol.

General comments

This paper is written precisely, logically and clearly. It presents novel methodology and scientific results that are very useful for atmospheric scientists, both experimentalists and modelers. I think this paper should be published in ACP after minor revision.

We thank the Anonymous Referee for his/her time in reviewing the manuscript and appreciate the constructive comments.

Specific comments

1. Page 1, Abstract; line 24; "simplistic inorganics apportionment methods" is unclear to me, do you mean PMF/ME2?

Specifically this refers to ion balance schemes described in 2.4 and the comparison discussed in 3.3.1. We changed this to: *"Compared to traditional, ion balance based inorganics apportionment schemes for aerosol mass spectrometer data, […]"*

2. Page 4, CToF-AMS; what is the mass resolving power of CToF? Is it possible to use high resolution mass spectra instead of UMR-MS? How reliably you can identify alkalimetals, especially rubidium, with CToF?

Mass resolution for our instrument was around 500 (*m/dm*), so it's not strictly high resolution capable. However, the resolution is enough to confirm presence of ions when distance between peaks is high enough, such as for some inorganics and metals with clearly negative mass defect. *m/z* calibrated mass spectrum below for *m/z* 80 to 91 Th is shown below. In addition to the natural isotopic ratio mentioned in 3.4.1, the exact masses for [85]Rb (84.912 a.m.u.) and [87]Rb (86.909 a.m.u.) correspond to the measured spectrum, supporting the Rubidium interpretation. We added the figure to the supplementary material (Fig. S.11) and a reference to Sect 3.4.2.

[Figure]

3. Page 5, Section 2.2.2 Data preparation and down-weighting; Could you explain here (shortly) how RIEs and fragmentation table were taken into account?

RIE and fragtable are (shortly) explained in the previous section (Sect 2.2.1; "The aerosol mass spectrometer (AMS) instrument and basic data processing"): "*The per-amu (atomic mass unit) analyser signal is subsequently quantified based on instrument response calibrations and corrections (among others the correction for relative ionisation efficiency between the species; RIE; Allan et al., 2004); supplementary information Sect S.4). Individual, unit-mass-resolution amu signals are then chemically speciated, based on chemical information on fragmentation and air composition (see Allan et al., 2003b), for details)*". To maintain consistency of the level of technical specifics discussed in the article, we would prefer to avoid very detailed explanations on individual technical topics such as (R)IE calibration/correction or the inner workings of the fragtable. For an interested reader, a detailed description on these topics can be found in the Allan *et al.* papers, as referenced.

4. Page 6, line 28-29, "all the source profiles are constrained, but allowed to vary within narrow limits", what alpha-value (constraning value)?

See similar comment by Anonymous Referee #1. We added here:
"*[…] allowed to vary within narrow limits (derived from variability estimates; see Sect 2.5)*". We also added a sentence on this to 2.5, and added a figure to Supplementary material depicting the final variability estimate (P-III; Figure S.13).

[Figure]

5. Page 11, line 1; "we applied an ion ratio Rcalib = 0.42, taken as the average of mass spectrum based AN calibrations (S.I Sect S.6)". Do you mean m/z 46/30 in Fig. S3? It seems to be much larger than 0.42. In general, Figure S3 is very difficult to read because it is unclear and has very small fonts. Please improve the quality of the figure.

*The value 0.42 is from an older IE calibration analysis we did not have available in full for this work. As stated in the caption, the MS-mode calibration data in Figure S.3. is unprocessed (meaning incomplete in terms of fragtable corrections for m/z 30 Th etc., as it was omitted from the main analysis), so it was not used quantitatively. We agree with the Referee that the quality of Figure S.3. in supporting material was poor. The figure was re-drawn.*

[Figure]

6. Page 15, line 16-18; similar comment, for cluster #8 m/z 46/30=0.44, according to Figure 3 the ratio is larger

*The ratio mentioned in text (0.44) is taken from the final (P-III) result (Figure 4), as are all similar diagnostics values. We tried to clearly state this in the beginning of results section (Sect. 3., p.13): "For easier comparability, all ratios and fractions of signals presented in the following sections are similarly calculated from the corresponding final spectra (P-III)." We are aware that our decision to report all diagnostics as values from the final model (P-III) may be a source of some confusion. However, given the multi-phase methodology, it would likely be still more confusing to report all the values from different phases / solutions. The purpose of the values is also to provide values for reference for future studies, so we think reporting (only) P-III values is the clearest (or at least the least confusing) way. In order to further decrease the risk of confusion, we added a notice after the p-15 l.16-18 sentence: "We note once more that these characteristic values for clusters are from the final model (P-III; Figure 4), as outlined before."*

7. Page 30, line 12, what is the origin of KNO3 in Hyytiälä?

This has not been investigated for Hyytiälä specifically. We assume the main source of submicron K for Hyytiälä be to be biomass burning. e.g. (Li et al., 2003). We added this mention to Sect 3.2.2 (p.23, l.21).

8. Results and discussion in general: diurnal trends of the factors are not utilized at all when interpreting and identifying aerosol components, and similarly, any auxiliary gas (or particle) data is not exploited. Couldn't this additional data help interpreting the results?

We added a figure of the components' diurnal behaviour to supplementary material. However, due to the varying conditions of long-distance transport of aerosol to the site, the diurnal profiles are not as useful as for more urban sites. Thus, the uncertainties of the information in diurnal profiles are high. There is also plenty of auxiliary data available, and it is true this data is not fully exploited in this work. However, one of the main points of this work is to base most of the analysis on statistical diagnostics and machine learning methods, and focus less on auxiliary data and traditional PMF evaluation criteria, such as correlations with trace gases and examining diurnal cycles. We find that identifying the main components is rather unambiguous from the spectral similarities alone, and a detailed examination / interpretation of the outliers is somewhat outside of the scope of this work due to the already large volume of data and methods presented. We do agree there is a lot of room for a more detailed study of the auxiliary data, diurnal cycles and more in-depth interpretation of the chemical processes and source attribution etc., and would encourage taking this up as a topic in future studies. We added the following to Sect. 3.1.3:
*"Diurnal cycles of the components for the entirety of data are available in S.I (Figure S.12). Due to the rural setting of the site and the generally long transport times of aerosol before reaching the site, diurnal cycles for the various aerosol types are not as characteristic as they would be for urban measurements (for e.g. temporal trends of HOA and BBOA). Also due to seasonal differences, the variability between data sets is considerable, resulting in high uncertainty in interpretation. The daily cycles are likely a mixed product of source emissions, boundary layer dynamics and aerosol temperature response. While of interest, disentangling these processes is beyond the topic of this study."*

9. Supplemental material, Figure S5; could you add total mass for each data set?

Good suggestion. As the campaign specific absolute mass loadings may be of interest, we added a panel to Figure S.5. with this information.

[Figure]

Technical corrections:

10. Page 11, line 18-19, second parenthesis is missing
Corrected.

11. Page 30, line 11; non-quantitative
Corrected.

**References**

Li, J., Pósfai, M., Hobbs, P. V., and Buseck, P. R. J. J. o. G. R. A.: Individual aerosol particles from biomass burning in southern Africa: 2, Compositions and aging of inorganic particles, 108, 2003.